# C-terminal binding protein 2 interacts with JUNB to control macrophage inflammation

Benjamin A Strickland[1], Antonia Babl[1], Lena Wolff[1], Priya Singh[2], Marika E Friano[1], Franziska Greulich[1], N Henriette Uhlenhaut[1,2]

**Although acute inflammatory responses are critical for survival, chronic inflammation is a leading cause of disease and mortality worldwide. Nevertheless, our mechanistic understanding of pathogenesis is still limited and precise treatment options are lacking. Here, we investigate the role of the transcriptional co-repressors C-terminal binding protein (CTBP) 1 and 2 in murine and human macrophage activation using loss-of-function models to show that CTBP2 but not CTBP1 controls inflammatory gene expression. We find that CTBP2 occupies *cis*-regulatory elements of inflammatory genes together with the transcription factors NF-κB and AP-1 and forms a co-repressor complex. Rescue of *Ctbp1/2* double knockout cells with WT, oligomeric CTBP2 attenuates inflammatory responses, whereas a monomeric mutant does not. Differential profiling of CTBP2's WT and monomeric interactome confirms oligomer-specific interactions with multiple repressors. Conversely, monomers retain the ability to interact with AP-1 and RNA polymerase II, boosting gene expression. Our findings point to an important function for CTBP2 in fine-tuning inflammatory gene expression, potentially unveiling novel therapeutic targets for the treatment of inflammatory diseases.**

## Introduction

Inflammation is a tightly regulated response of the body to pathogenic signals or tissue damage. After initial sensing by innate immune cells such as macrophages, other immune cells are recruited by cytokines such as interleukins (e.g., IL1A, IL12B) or chemokines (e.g., CXCL1, CCL22) (Turner et al, 2014). Ultimately, this carefully balanced process aims to stop invading pathogens and clear cellular damage. However, in certain cases such as in inflammatory bowel disease and other autoimmune diseases, inflammatory reactions are dysregulated, leading to hyperactive immune cells and damage to the own body (Bach, 2002; Pisetsky, 2023). Although glucocorticoids are first-line immunosuppressive

drugs administered in the clinic, adverse side effects, including metabolic disturbances, severely hamper their applicability (Cain & Cidlowski, 2015; Strickland et al, 2022). Therefore, novel molecular drug targets for the treatment of inflammatory diseases are urgently required. A recent screen for co-regulators involved in transcriptional regulation in inflammatory macrophages upon glucocorticoid treatment identified C-terminal binding proteins (CtBPs) as immunomodulating co-regulators interacting with the glucocorticoid receptor (Greulich et al, 2021).

CtBPs are highly conserved transcriptional co-repressors that are described to regulate multiple aspects of transcription, including histone modification, histone positioning, and post-translational modification of transcription factors (Ray et al, 2014; Kim et al, 2015; Bi et al, 2022). Especially, writers and readers of histone H3 tail modifications are known to interact with CtBPs and are involved in mediating their transcriptional actions (Kuppuswamy et al, 2008). In mammals, CtBPs are represented by the two closely related family members CTBP1 and CTBP2, which are described as largely redundant transcriptional repressors (Stankiewicz et al, 2014). In the context of inflammation, evidence suggests that both CTBP1 and CTBP2 can repress NF-κB activity in luciferase reporter assays and down-regulate inflammatory gene expression in macrophages (Shen et al, 2017). In contrast, other studies attribute pro-inflammatory functions to CtBPs (Li et al, 2020, 2022). Altogether, CtBPs have been suggested to regulate inflammatory responses; however, their molecular mechanisms remain elusive (Shen et al, 2017).

Among transcriptional co-regulators, CtBPs are of special interest because they uniquely display dehydrogenase activity and possess a NAD(H)- and a substrate-binding site (Kumar et al, 2002). Despite their unresolved enzymatic function, the binding of NAD(H) triggers a conformational change in CtBPs, which strongly fosters self-association to dimers and tetramers, thereby potentially conveying metabolic information (Kumar et al, 2002; Zhang et al, 2002; Fjeld et al, 2003; Bellesis et al, 2018; Jecrois et al, 2021; Nichols et al, 2021; Erlandsen et al, 2022). More recently, CTBP2 was shown to bind Acyl-CoA, causing impaired di- and oligomerization (Sekiya et al, 2021, 2023; Saito et al, 2023). Together, this indicates that CtBPs

[1]Metabolic Programming, TUM School of Life Sciences, ZIEL-Institute for Food and Health, Technical University of Munich, Freising, Germany  [2]Institute for Diabetes and Endocrinology (IDE), Helmholtz Munich (HMGU) and German Center for Diabetes Research (DZD), Neuherberg, Germany

Correspondence: franziska.greulich@tum.de; henriette.uhlenhaut@tum.de

may act as sensors of cellular metabolism via metabolite-induced alterations in their oligomeric state. Mechanistically, distinct oligomeric states are associated with different transcriptional outcomes, integrating metabolic information into gene regulation (Bhambhani et al, 2011). In mammals, both gene-activating and gene-repressing functions are described to require CtBP oligomers (Ray et al, 2017; Jecrois et al, 2021). Exploration of CtBP oligomerization mutants revealed a loss of target gene repression and reduced interaction with other transcriptional regulators in human cancer cell lines (Kumar et al, 2002; Jecrois et al, 2021; Sekiya et al, 2021; Li et al, 2023). Conversely, the importance of oligomer formation for transcriptional regulation by CtBPs during macrophage inflammation has not been investigated yet.

Here, we show that CTBP2 but not CTBP1 is a transcriptional co-repressor of inflammatory responses in macrophages. We demonstrate that CTBP2 occupies *cis*-regulatory elements of inflammatory genes, alongside the inflammatory transcription factor NF-κB and the AP-1 family member JUNB. CTBP2 physically interacts with these transcription factors, and multiple transcriptional co-repressors in inflammatory macrophages. With loss- and gain-of-function studies using the macrophage-like cell line J774.1, we validate that CTBP2 regulates inflammatory gene expression. Oligomerization-defective mutants of CTBP2 fail to limit inflammatory gene expression and lose interaction with multiple co-repressors including WIZ and KDM1A but not JUNB. We suggest that CTBP2 physically interacts with JUNB and, depending on CTBP2's oligomeric state, bridges it with transcriptional repressors, diminishing inflammatory gene expression.

## Results

### CTBP2 blunts inflammatory responses in macrophages

To understand the role of CtBPs during inflammatory responses in macrophages, we knocked down *Ctbp1* or *Ctbp2* in murine BMDMs treated with either vehicle control or LPS—a bacterial toxin stimulating Toll-like receptor 4 signaling (Fig 1A). Subsequent gene expression profiling by RNA-seq revealed 69 differentially expressed genes (baseMean > 100, fold change > 1.4, *P* < 0.05) after knockdown of *Ctbp1* and 158 differentially expressed genes after knockdown of *Ctbp2* indicating a dominant gene regulatory role of *Ctbp2* in macrophages upon LPS stimulation (Fig S1A and B). Gene ontology enrichment analysis of differentially expressed genes for "Biological Process" revealed that upon knockdown of *Ctbp2*, genes involved in "cell chemotaxis" and "cytokine-mediated signaling pathway" were up-regulated in the LPS condition (Fig 1B). In addition, we also observed down-regulation of genes controlling "nuclear division" after *Ctbp2* knockdown, which is in line with the reported role of CTBP2 in promoting cell cycle progression in numerous malignancies (Wang et al, 2015; Dai et al, 2017; Zhao et al, 2019) (Fig S1C). Of note, most of the LPS-responsive genes were not affected by knockdown of *Ctbp1* or *Ctbp2*, indicating a specific role of *Ctbp2* in fine-tuning a subset of inflammatory genes upon LPS challenge (Fig S1D). The expression of *Ctbp2*-dependent genes associated with "cell chemotaxis" and "cytokine-mediated signaling pathway" shown as a heatmap highlighted that *Ctbp2* but not *Ctbp1* suppresses inflammatory gene

expression (Fig 1C). We confirmed that *Ctbp1* and *Ctbp2* have nonredundant roles in controlling macrophage inflammatory responses in an independent knockdown experiment followed by RT–qPCR for the selected target genes *Il1a*, *Ccl22*, and *Il12b* (Figs 1D and S1E). In line with the RNA-seq experiments, loss of *Ctbp2* induced hyperactivation of those pro-inflammatory cytokines upon LPS stimulation. To address whether *Ctbp2*'s role in controlling macrophage inflammation is conserved in human, we performed *CTBP1* and *CTBP2* knockdowns in human monocyte-derived macrophages. Indeed, knockdown of *CTBP2* but not *CTBP1* led to a strong trend of elevated *CCL22* and *IL12B* expression after LPS treatment (Figs 1E and S1F). However, regulation of *IL1A* was not dependent on *CTBP2* alone, potentially indicating redundant roles of *CTBP1* and *CTBP2* in human monocyte-derived macrophages.

### CTBP2 occupies JUNB and RELA DNA motifs close to inflammatory genes in macrophages

Because we found that CTBP2 specifically fine-tunes the transcriptional response to LPS in macrophages, we asked whether CTBP2's effects on inflammatory gene expression are mediated directly by enhancer interactions using ChIP-seq in murine BMDMs treated with either vehicle or LPS. We observed 20,685 CTBP2-occupied sites shared between vehicle and LPS conditions, whereas 2019 sites were gained and 683 sites were lost after LPS treatment (Fig 2A). Most of the shared sites were located in enhancers (intergenic or intronic), whereas only a small proportion was localized to promoters. This indicates that CTBP2 is mainly bound to enhancers, a binding site distribution also reflected in the LPS- and vehicle-specific subsets (Fig 2B). Because we only observed a minor redistribution of chromatin-bound CTBP2 upon LPS stimulation, we performed differential binding analysis to probe for changes in CTBP2 occupancy upon LPS stimulation. Differential binding analysis revealed that LPS decreased CTBP2 occupancy at 101 genomic loci and increased CTBP2 occupancy at 1926 loci—among them many in proximity to genes with differential expression upon *Ctbp2* knockdown such as the pro-inflammatory cytokines *Il1b*, *Cxcl1*, and *Ccl2* (Figs 2C and S1B). Gene ontology enrichment for "Biological Process" of genes in proximity to CTBP2-occupied sites confirmed that LPS-gained sites were enriched for "positive regulation of defense response" and "cytokine-mediated signaling pathway" (Fig S2A). In contrast, genes near unaffected sites were enriched for housekeeping functions such as "mRNA processing" or "histone modification" and genes near lost sites were enriched for "negative regulation of hydrolase activity" with low significance (Fig S2B and C). Similar to *Ctbp2* knockdown experiments (Fig 1B and C), this suggests that CTBP2 controls inflammatory gene expression programs in response to LPS. Motif enrichment analysis at differentially occupied loci demonstrated that CTBP2 occupancy was gained at RELA and JUNB DNA motifs in response to LPS, suggesting that these transcription factors may recruit CTBP2 to the respective sites (Fig 2D). In particular, we observed increased CTBP2 occupancy after LPS treatment in proximity to the CTBP2 target genes *Il1a*, *Ccl22*, and *Il12b* (Figs 2E and S2D). Overlay with public ChIP-seq data for JUNB (GSE38379 [Ostuni et al, 2013]) and RELA (GSE16723 [Barish et al, 2010]) in LPS-treated BMDMs indicates that CTBP2 was recruited to sites also bound by

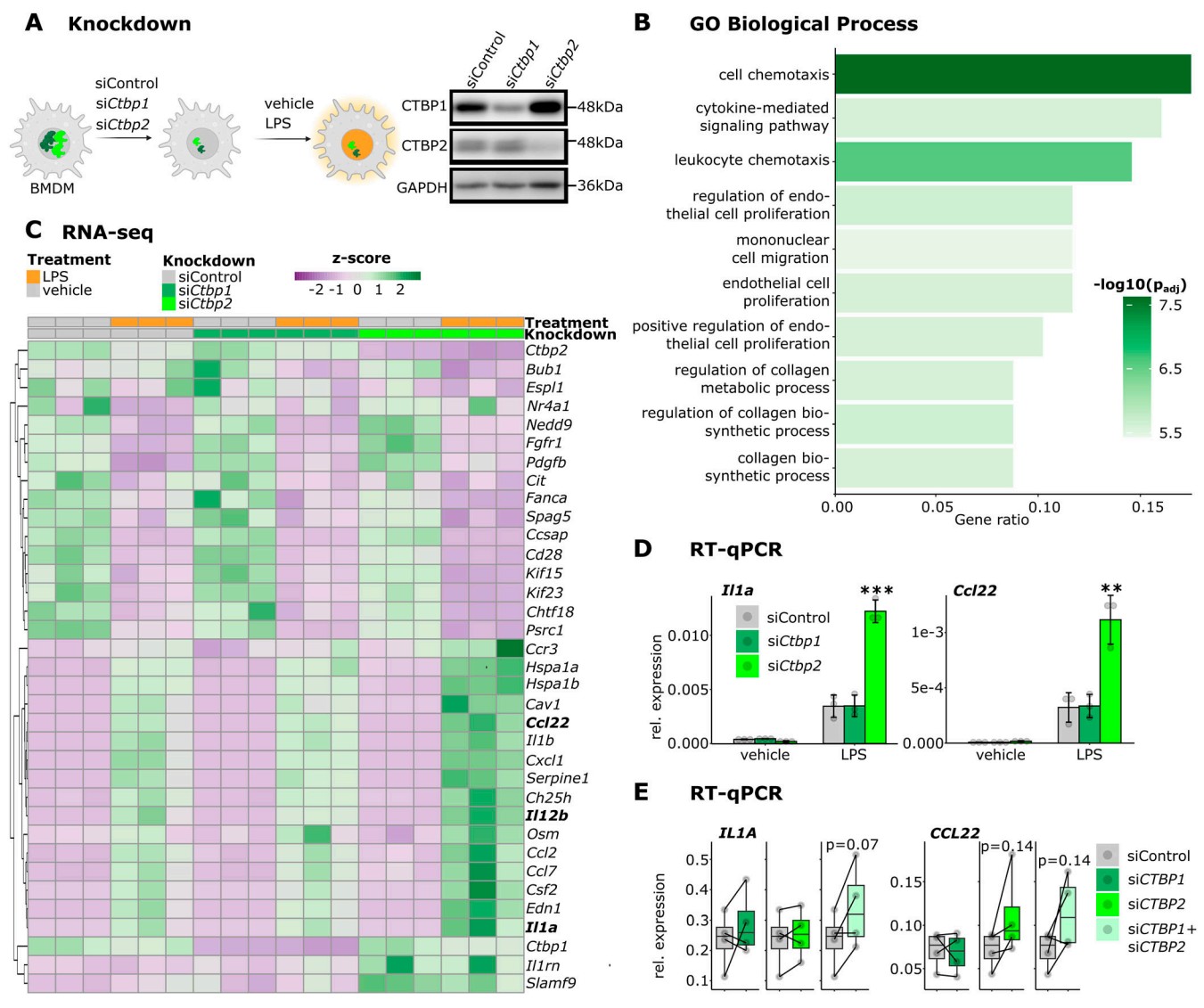

**Figure 1. CTBP2 limits inflammatory gene expression in macrophages.**
**(A)** Experimental overview and Western blot of CTBP1, CTBP2, and GAPDH (loading control) after knockdown of *Ctbp1* and *Ctbp2* in BMDMs. Representative example of n = 3 biological replicates. **(B)** Gene ontology (GO) enrichment for "Biological Process" in CTBP2-repressed genes (fold change > 1.4, *P* < 0.05 after *Ctbp2* knockdown, Fig S1B). Color intensity indicates $\log_{10}$-transformed Benjamini-Hochberg adjusted *P*-value. **(C)** Heatmap displaying RNA-seq expression (z-score of TPMs) of genes from GO terms "cell chemotaxis," "cytokine-mediated signaling pathway," and "nuclear division" with absolute fold change > 1.4 upon knockdown of *Ctbp2*. n = 3. **(D)** RT–qPCR experiments for *Il1a* and *Ccl22* after *Ctbp1* or *Ctbp2* knockdown in BMDMs. Relative expression over *Rplp0* from n = 3; gray dots represent individual data points, and error bars indicate SD. *$P$ < 0.05, **$P$ < 0.01, ***$P$ < 0.001, ANOVA followed by Tukey's test. **(E)** RT–qPCR experiments for *IL1A* and *CCL22* after *CTBP1*, *CTBP2*, or *CTBP1/2* double knockdown in LPS-stimulated human macrophages. Relative expression over *RPLP0* from n = 4, gray dots are individual data points, and error bars show SD. *P*-values are displayed above boxes, paired *t* tests.

these transcription factors specifically in the LPS condition. We validated the treatment-dependent association of CTBP2 with the *Il1a* promoter and a *Ccl22* enhancer by ChIP-qPCR and observed increased CTBP2 occupancy after LPS stimulation, corroborating our ChIP-seq results (Fig S2E).

### CTBP2 interacts with JUNB and multiple repressors in macrophages

To identify mediators of CTBP2-dependent transcriptional repression, we performed chromatin immunoprecipitation followed

by mass spectrometry (ChIP-MS) for CTBP2 in BMDMs treated with LPS. This confirmed that CTBP2 associated with inflammatory transcription factors including NF-κB family members (NFKBIZ, NFKB1, RELA) and the AP-1 family member JUNB (Fig 3A). In addition, we observed enrichment of RNA polymerase II subunits, the NuRD complex, and other known transcriptional repressors such as BCOR and ETV3 (Figs 3A and S3A). We also identified previously described CTBP2 interactors including ZFP217, WIZ, and KDM1A (Hildebrand & Soriano, 2002; Quinlan et al, 2006; Ueda et al, 2006; Kuppuswamy et al, 2008; Ray et al, 2014). This interactome study positions CTBP2 at the interface of pro-inflammatory transcription factors, RNA

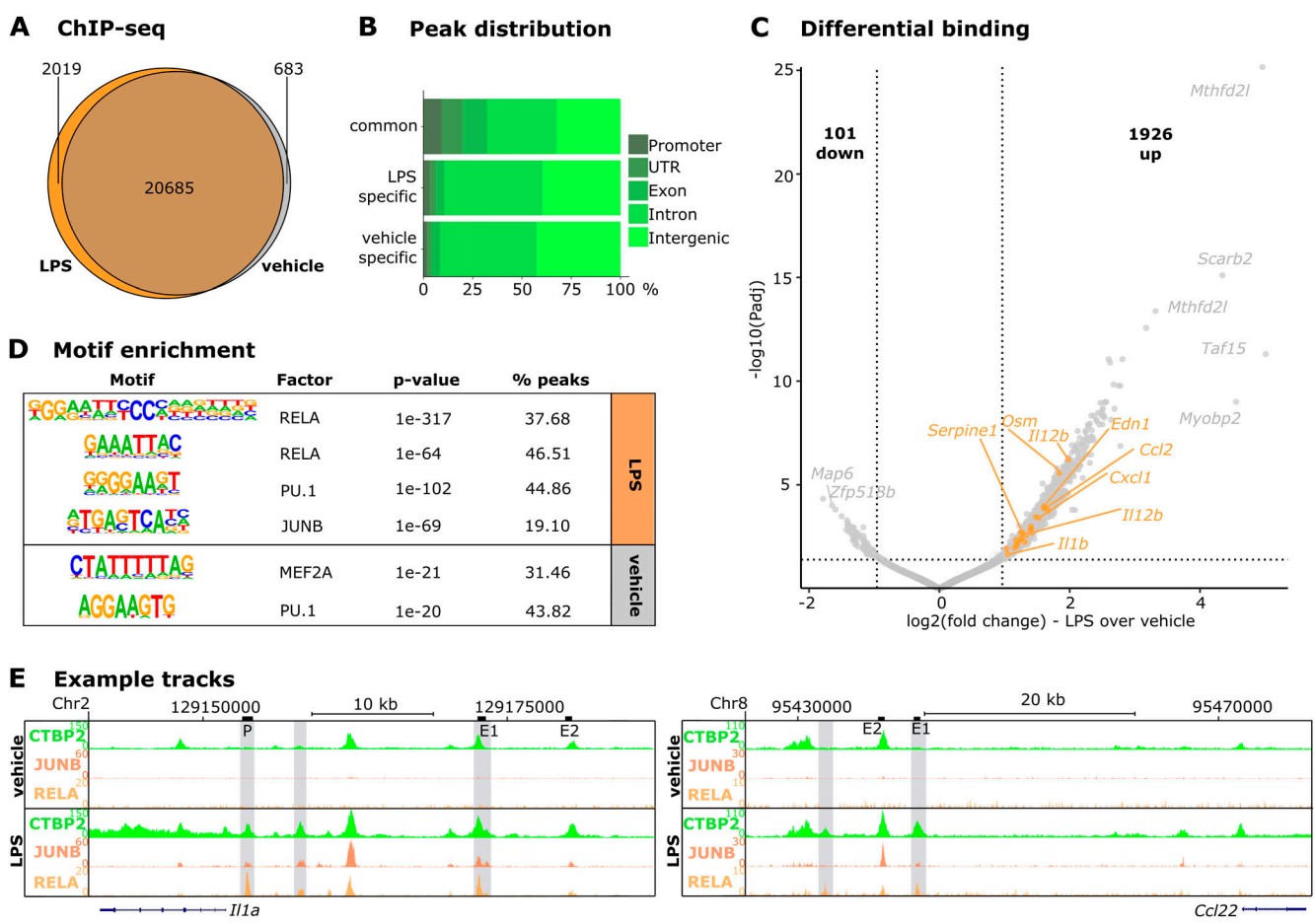

**Figure 2. CTBP2 occupies JUNB and RELA DNA motifs close to inflammatory genes in macrophages.**
**(A)** Venn diagram of CTBP2-occupied loci in LPS and vehicle conditions. Numbers indicate the total reproducible binding sites in BMDMs determined by ChIP-seq. n = 3. **(B)** Distribution of CTBP2-occupied sites to indicated genomic positions separated by subsets from (A). **(C)** ChIP-seq volcano plot of CTBP2 differential occupancy in LPS- versus vehicle-stimulated BMDMs. Solid dots are differential peaks (absolute fold change > 2 and $P$adj < 0.05), and orange dots indicate peaks annotated to genes regulated by CTBP2 in BMDMs and gene ontology terms "cytokine-mediated signaling pathway" and "cell chemotaxis" in Fig S1B. Bold numbers indicate the amount of differentially occupied sites. n = 3. **(D)** Motif enrichment of differentially CTBP2-occupied sites from C in vehicle and LPS conditions. Indicated is the transcription factor with the closest known motif, $P$-value, and motif abundance in peaks. **(E)** Example ChIP-seq tracks of CTBP2 upstream of *Il1a* and *Ccl22* in vehicle and LPS conditions. Additive tracks from n = 3. The scale is displayed left next to the tracks. Tracks are shown together with public ChIP-seq data from JUNB (GSE38379 [Ostuni et al, 2013], n = 1) and RELA (GSE16723 [Barish et al, 2010], n = 1 [vehicle] and n = 2 [LPS]). LPS-specific CTBP2 occupancy together with these transcription factors is highlighted in gray. Black bars over the tracks indicate the primer positions for ChIP-qPCRs in Figs 3E and S2E. Locus information is indicated above the tracks.

polymerase II, chromatin modifiers, and transcriptional repressors in macrophages. To analyze whether LPS treatment can alter CTBP2's interactome, we performed differential ChIP-MS between LPS- and vehicle-treated BMDMs (Figs 3B and S3B). Upon LPS stimulation, CTBP2 specifically interacted with 17 proteins including the inflammatory transcription factor JUNB, multiple NF-κB family members (NFKBIZ, NFKB1, RELA), and the repressors BCOR and ETV3. Especially, the NF-κB transcription factor family members gain nuclear localization and DNA binding upon LPS stimulation (Oeckinghaus et al, 2011; Ernst et al, 2018), which may make them available for the interaction with CTBP2. Conversely, CTBP2's interaction with other proteins such as ZNP217, WIZ, KDM1A, and the NuRD complex was treatment-independent. The strong gain in JUNB interaction upon LPS treatment correlates with the LPS-specific JUNB binding at CTBP2-occupied sites observed by ChIP-seq (Fig 2E). Of note, the interaction of CTBP2 with the closely related AP-1 family

member JUN, which was previously shown to interact with CTBP2 (Chen et al, 2022), was diminished after LPS stimulation (Fig 3B). From those observations, we suggest a stimulus-dependent redistribution of co-regulators between AP-1 family members. Gene ontology enrichment for "Molecular Function" highlighted the association of CTBP2 with proteins showing "transcriptional co-repressor activity" and "RNA polymerase II–specific DNA-binding transcription factor binding" (Fig 3C). Among the LPS-specific interactors, gene ontology enrichment for "Molecular Function" further stressed regulation at the "RNA polymerase II core promoter" with "RNA polymerase II–specific DNA-binding transcription repressor activity," indicating that CTBP2 protein interactions at gene regulatory elements may be required to control target gene expression (Fig S3C). We next validated the interaction of CTBP2 with inflammatory transcription factors performing NanoBRET assays (Machleidt et al, 2015) in J774.1 macrophages after LPS stimulation.

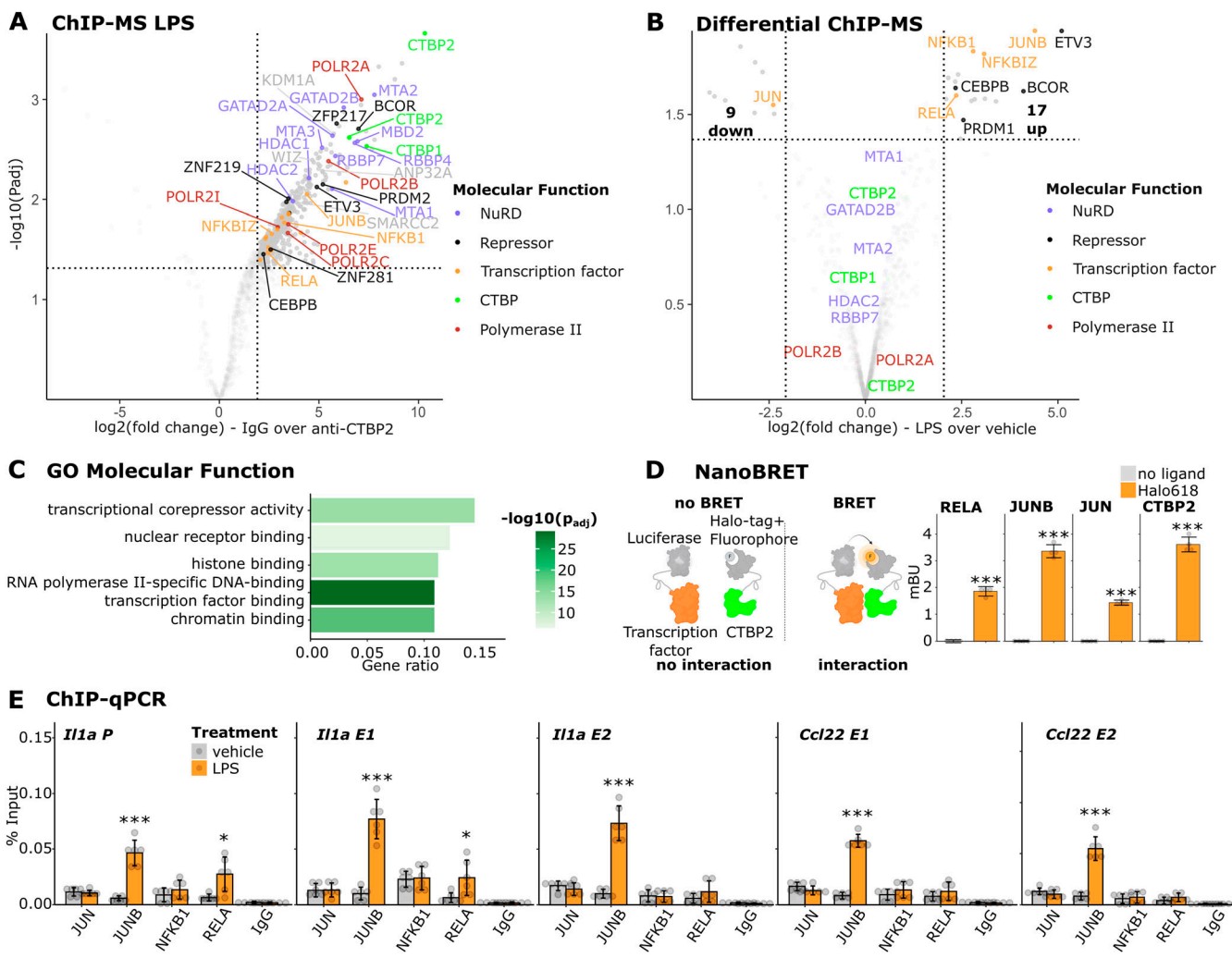

**Figure 3. CTBP2 interacts with JUNB, NF-κB, and multiple repressors in inflammatory BMDMs.**
**(A)** ChIP-MS volcano plot in LPS-treated macrophages. Solid dots are significantly enriched proteins over IgG control (fold change > 4 and Padj < 0.05). Colors indicate association with specific molecular functions (see legend). n = 3. **(B)** Volcano plot of differential ChIP-MS interactors of CTBP2 in LPS versus vehicle. Solid dots are significantly altered protein interactions (absolute fold change > 4 and Padj < 0.05). Colors indicate the molecular function of differential interactors (see legend). n = 3. **(C)** Gene ontology (GO) enrichment for "Molecular Function" of LPS-specific CTBP2 interactors in BMDMs. Color intensity indicates log₁₀-transformed Benjamini-Hochberg adjusted P-value. **(D)** NanoBRET assay in LPS-treated J774.1 cells expressing CTBP2-Halo-tag and indicated proteins fused to Nano-luciferase. Bioluminescence resonance energy transfer in milliBRET units (mBU). n = 4 technical replicates. Gray dots are individual data points, and error bars show SD. ***P < 0.001, two-sided t test. **(E)** ChIP-qPCR experiments for JUN, JUNB, NFKB1, and RELA at CTBP2-bound loci in BMDMs. IgG was used to monitor unspecific binding. Primer positions are indicated by black bars in Fig 2E. Percent input from n = 3. Treatment is indicated by color. Gray dots are individual data points, and error bars show SD. *P < 0.05, ***P < 0.001, two-sided t test.

Bioluminescence resonance energy transfer from Nano-luciferase–labeled RELA, Nano-luciferase–labeled JUNB, or Nano-luciferase–labeled JUN to fluorescently labeled CTBP2-Halo-tag was measured (Fig 3D). Although this demonstrates the ability of CTBP2 to physically interact with these transcription factors, JUNB outperformed RELA and JUN, suggesting that JUNB might have a higher affinity for CTBP2 compared with RELA and JUN. Because CTBP2 is known to assemble into di- and tetramers (Kumar et al, 2002; Bellesis et al, 2018; Jecrois et al, 2021), we used Nano-luciferase–labeled CTBP2 as a positive control. Finally, we tested stimulus-dependent DNA binding of NF-κB and AP-1 family members, which we found to interact with CTBP2 at LPS-gained CTBP2 binding sites (Fig 2E). ChIP-qPCR experiments validated

the recruitment of JUNB to CTBP2-occupied *cis*-regulatory elements near *Il1a* and *Ccl22*, whereas RELA was only recruited to the *Il1a* promoter and one out of two *Il1a* enhancers, but not to the *Ccl22* enhancers (Fig 3E). We did not observe binding of NFKB1 nor JUN at any investigated site (Fig 3E).

## CTBP2 blunts inflammatory responses mediated by NF-κB and AP-1 in J774.1 macrophages

To investigate the mechanisms of CtBP-mediated inflammatory gene repression, we used the macrophage-like cell line J774.1 and generated single and double knockouts of *Ctbp1* and *Ctbp2* (here: *Ctbp1* KO, *Ctbp2* KO, *Ctbp* dKO) using CRISPR/Cas9 (Fig S4A). The

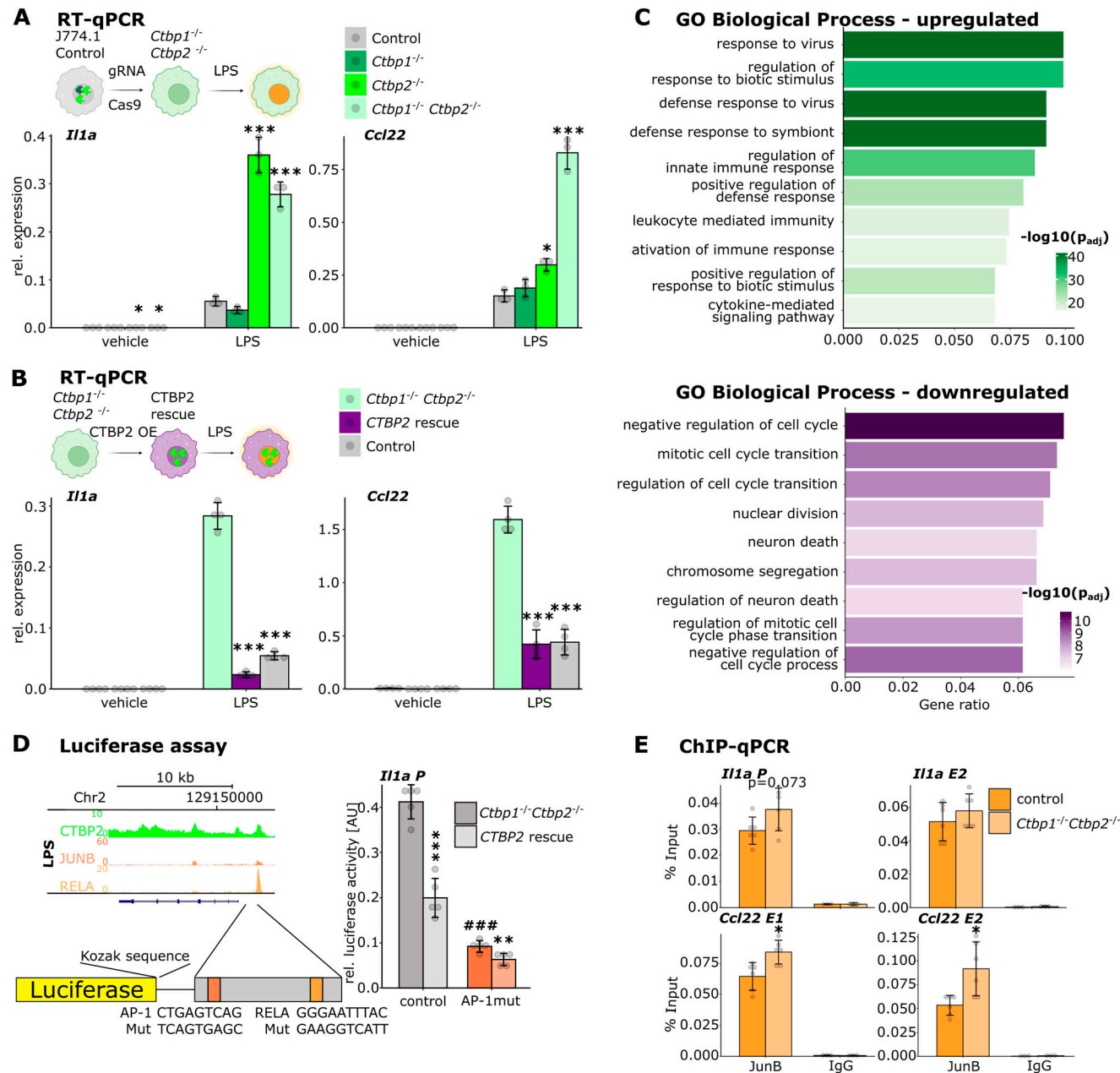

**Figure 4. CTBP2 represses inflammatory gene expression in J774.1 macrophages.**
**(A)** RT–qPCR in J774.1 cells for *Il1a* and *Ccl22* after *Ctbp1*, *Ctbp2*, or *Ctbp1/2* double knockout (*Ctbp* dKO). Relative expression of the indicated gene over *Rplp0* from n = 3. Colors indicate the genotype. Gray dots are individual data points, and error bars show SD. *P < 0.05, ***P < 0.001, ANOVA followed by Tukey's test. **(B)** RT–qPCR in J774.1 cells for *Il1a* and *Ccl22* in *Ctbp* dKO cells stably transfected with empty vector (Control) or *CTBP2* expression vector (*CTBP2* rescue) and unmodified WT cells. Relative expression of the indicated genes over *Rplp0* from n = 3. Colors indicate the genotype. Gray dots are individual data points, and error bars show SD. *P < 0.05, ***P < 0.001, ANOVA followed by Tukey's test. **(C)** Gene ontology (GO) enrichment analysis for "Biological Process" of up-regulated genes (green, fold change >1.4 and FDR < 0.05, Fig S4D) and down-regulated genes (purple, fold change < –1.4 and FDR < 0.05, Fig S4D) in J774.1 *Ctbp* dKO cells compared with WT cells upon LPS stimulation. Color intensity indicates log$_{10}$-transformed Benjamini-Hochberg adjusted *P*-value. **(D)** Luciferase assay using an *Il1a* promoter element with one AP-1 and one NF-κB binding site to drive luciferase reporter expression in J774.1 *Ctbp* dKO cells. Relative luciferase activity with or without transient CTBP2 expression. n = 4 technical replicates. The binding site for AP-1 was mutated as shown on the x-axis. Gray dots are single data points, and error bars show SD. **P < 0.01, ***, ###P < 0.001, two-sided *t* test between *CTBP2*-expressing and non-expressing cells (*) or between control sequence and sequence with mutation in the AP-1 binding site (#). **(E)** ChIP-qPCR for JUNB at CTBP2-bound loci in J774.1 WT or *Ctbp* dKO cells after LPS treatment near CTBP2-regulated inflammatory genes, as indicated by black bars in Fig 2E. Percent input from n = 3 replicates with technical duplicates each. *P*-value is indicated above bars. *P < 0.05, two-sided *t* test.

knockout was confirmed by Western blot and immunostaining (Fig S4B and C). RT–qPCR experiments confirmed that the loss of *Ctbp2*, but not of *Ctbp1*, caused the elevated expression of *Il1a* and *Ccl22* in J774.1 cells upon LPS stimulation (Fig 4A), similar to primary macrophages (Fig 1D). Analyzing the expression kinetics of these genes in WT and *Ctbp* dKO cells over an extended time course of 12 h

showed that loss of CtBPs elevates inflammatory responses at every investigated time point, without altering expression kinetics (Fig S4D). Of note, the expression of anti-inflammatory mediators such as *Dusp1* and *Tsc22d3* is not lost in *Ctbp* dKO cells (Fig S4E). In line with those observations, the stable re-expression of CTBP2 in *Ctbp* dKO cells was sufficient to suppress the hyperactivation of *Il1a* and *Ccl22* upon LPS treatment (Fig 4B).

In addition, we profiled global transcriptomic changes in *Ctbp* dKO cells upon LPS stimulation by RNA-seq. Similar to our observations in knockdown experiments in BMDMs (Fig 1C), *Ctbp* dKO cells were prone to elevated inflammatory gene expression in response to LPS with multiple interleukins and chemokines being up-regulated compared with control cells (Fig S4F). Gene ontology enrichment for "Biological Process" underlined the requirement of CtBPs in the suppression of inflammation. In contrast, genes with lost expression in *Ctbp* dKO cells were associated with "nuclear division," similar to the *Ctbp2* knockdown in BMDMs (Figs 4C and S1C).

After establishing that J774.1 *Ctbp* dKO cells recapitulate the *Ctbp* loss-of-function phenotype of primary macrophages, we investigated whether CTBP2 directly controls JUNB or RELA-mediated transcription. Therefore, we designed a luciferase reporter with a *cis*-regulatory element containing the *Il1a* promoter, for which we have shown CTBP2 (Fig 2), JUNB, and RELA binding in BMDMs upon LPS stimulation (Figs 3E and 4D). We selected this regulatory element because it controls the expression of a prominent CTBP2 target gene and contains only one NF-κB and one AP-1 motif. Transfection of J774.1 *Ctbp* dKO cells with the reporter construct resulted in high luciferase activity, indicative of prominent transcription upon CtBP loss. Luciferase activity was blunted upon the transient overexpression of WT CTBP2. Mutation of the single RELA binding site only weakly reduced luciferase expression, whereas mutation of the AP-1 binding site abolishes the activity of the luciferase construct, indicating that AP-1 and not NF-κB is the main driver of gene activation at the *Il1a* promoter (Figs 4D and S4G). CTBP2 re-expression was able to suppress transcriptional activity of the WT and RELA binding site-mutated reporter, emphasizing CTBP2's repressive potential independent of NF-κB binding. However, mutation of the AP-1 motif had already reduced the luciferase activity to a minimum in *Ctbp* dKO cells. Additional re-expression of CTBP2 only had a mild effect, indicating that AP-1 acts upstream of CTBP2 to control *Il1a* gene expression at the *Il1a* promoter (Fig 4D). Given the repressive effect of CTBP2 on AP-1–mediated transcription and the previously observed interaction with JUNB, we asked whether CTBP2 loss might affect JUNB binding to the DNA. ChIP-qPCR studies in WT and *Ctbp* dKO cells showed slightly increased JUNB binding to both *Ccl22* enhancers and a tendency of increased binding to the *Il1a* promoter (Fig 4E).

### CTBP2's oligomeric state influences inflammatory responses

CtBPs are known to fulfill distinct transcriptional functions depending on their oligomeric state (Bhambhani et al, 2011; Sekiya et al, 2021). To test whether CTBP2 oligomerization is required to limit JUNB activity, we investigated the impact of different CTBP2 oligomerization mutants on the expression of inflammatory genes. For this purpose, we stably re-expressed WT CTBP2 ("CTBP2 wt"), a

mutant defective of tetramerization (G216N, here "CTBP2 dim," Bellesis et al, 2018; Jecrois et al, 2021) and a mutant defective of dimerization (C140Y, N144R, R147E, L156W, here "CTBP2 mono," Kumar et al, 2002) in *Ctbp* dKO cells. All mutants showed nuclear localization and were expressed to similar amounts as in WT cells, whereas re-expressed CTBP2 wt was overexpressed (Fig S5A and B). As previously observed, rescue of *Ctbp* dKO cells with CTBP2 wt reversed the pro-inflammatory phenotype reducing transcript levels of *Il1a* and *Ccl22* after LPS stimulation. In contrast, CTBP2 dim only partially rescued *Ccl22* hyperactivation and failed to reduce *Il1a* expression. Surprisingly, CTBP2 mono activated expression of both *Il1a* and *Ccl22* even further (Fig 5A). Re-expression of the same mutants in *Ctbp2* single knockout cells confirmed these defects in gene regulation (Fig S5C). However, in contrast to *Ctbp* dKO cells, the CTBP2 mono mutant did not enhance inflammatory gene expression indicating a role of *Ctbp1*. Because CTBP2 mono failed to attenuate inflammatory gene expression, we investigated whether this mutant shows an altered interactome compared with CTBP2 wt. Therefore, we performed ChIP-MS for CTBP2 after LPS stimulation in J774.1 *Ctbp* dKO cells re-expressing either CTBP2 wt or CTBP2 mono. CTBP2 wt interacted specifically with components of the NuRD complex and the repressors KDM1A and WIZ, whereas CTBP2 mono lost those interactions. However, CTBP2 mono retained interactions with other proteins such as the chromatin remodeler SMARCC2 and the histone chaperone ANP32A. Moreover, it showed increased interactions with subunits of RNA polymerase II, in line with CTBP2 mono's behavior as a transcriptional activator (Figs 5B and S5D). Gene ontology enrichment analysis for "Molecular Function" highlighted that wild type but not monomeric CTBP2 interacts with proteins displaying "promoter-specific chromatin binding" and "transcriptional co-repressor activity" (Fig 5C). This indicates that CTBP2 oligomers may provide a scaffold for connecting transcription factors such as RELA and JUNB with co-repressors. Conversely, gene ontology enrichment analysis for "Molecular Function" of CTBP2 mono interaction partners showed that they were associated with "RNA polymerase II activity" (Fig 5D). In addition, CTBP2 mono interacted specifically with proteins displaying "acetyl-CoA C-myristoyltransferase activity" and "acetyl-CoA oxidase activity." Of note, while weakened, CTBP2 mono retained the interaction with JUNB as confirmed by NanoBRET in J774.1 cells (Fig 5E), suggesting that higher order oligomeric states of CTBP2 are not required for transcription factor interaction, but rather for co-repressor recruitment.

## Discussion

Our findings demonstrate that inflammatory gene expression upon LPS challenge is negatively regulated by CTBP2 but not CTBP1 in both murine and human macrophages (Fig 1). Similarly, previous studies in microglia also suggest CtBPs as repressors of inflammation (Saijo et al, 2011; Zhang et al, 2012; Shen et al, 2017). More specifically, these studies implicate CtBPs as co-repressors of the inflammatory transcription factors NF-κB (Shen et al, 2017) and c-Fos (Saijo et al, 2011); however, whether CTBP1 or CTBP2 is the key regulator of inflammatory programs has not been comprehensively

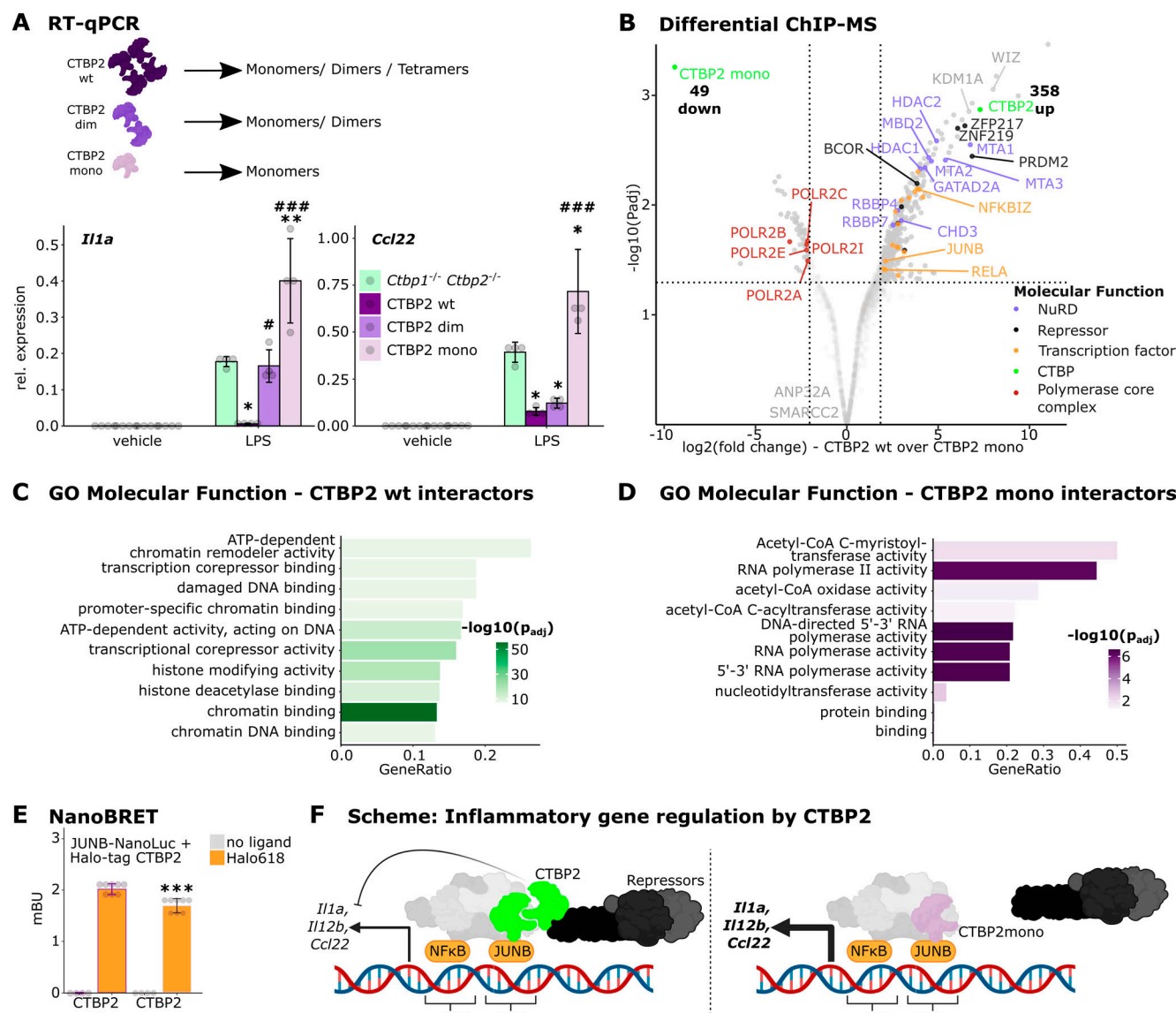

**Figure 5. CTBP2's oligomeric state influences repressor function by altering protein interactions.**
**(A)** RT–qPCR for *Il1a* and *Ccl22* in J774.1 *Ctbp1/2* double knockout (*Ctbp* dKO) cells stably expressing control vector or vectors encoding CTBP2 wt or CTBP2 mutants defective of oligomerization (CTBP2 dim and CTBP2 mono). Relative expression of indicated genes over *Rplp0* from n = 3 technical replicates. Gray dots are individual data points, and error bars show SD. *, #*P* < 0.05, ***P* < 0.01, ###*P* < 0.001, ANOVA followed by Tukey's test comparing with *Ctbp* dKO cells (*) or with *Ctbp* dKO cells re-expressing CTBP2 wt (#). **(B)** Volcano plot from ChIP-MS experiments in LPS-treated J774.1, comparing CTBP2 wt with CTBP2 mono–expressing cells. Solid dots are proteins significantly enriched in CTBP2 mono or CTBP2 wt (fold change > 4 and *P*adj < 0.05). Colors indicate association with distinct molecular functions (see legend). Bold numbers show the number of significant interactors of CTBP2 mono and CTBP2 wt. n = 3 technical replicates. **(C)** Gene ontology (GO) enrichment for "Molecular Function" of enriched CTBP2 wt–specific interactors in J774.1 cells in LPS condition. Color intensity indicates log₁₀-transformed Benjamini-Hochberg adjusted *P*-value. **(D)** GO enrichment for "Molecular Function" of CTBP2 mono–specific interactors in J774.1 cells after LPS treatment. Color intensity indicates the log₁₀-transformed Benjamini-Hochberg adjusted *P*-value. **(E)** NanoBRET assay in LPS-treated J774.1 *Ctbp* dKO cells transiently expressing CTBP2 wt-Halo-tag or CTBP2 mono-Halo-tag and Nano-luciferase–labeled JUNB. Bioluminescence resonance energy transfer in milliBRET units (mBU) with or without Halo-ligand 618 indicated by color. n = 4 (no ligand) and n = 8 (ligand) technical replicates. Gray dots are individual data points, and error bars show SD ***P* < 0.001, two-sided *t* test between CTBP2 wt and CTBP2 mono. **(F)** Schematic of CTBP2's gene regulatory role during macrophage inflammation. CTBP2 oligomers (green) interact with JUNB to recruit co-repressors and limit inflammatory gene expression. CTBP2 monomers (pink) retain JUNB interaction but cannot recruit co-repressors leading to increased inflammatory gene expression.

investigated. Our experiments reveal that in macrophages, CTBP2 restricts inflammatory responses, whereas CTBP1 is dispensable for the regulation of inflammatory gene expression upon LPS stimulation. This distinct role of CTBP2 in macrophage inflammation is surprising in light of the high homology between the two proteins

(Katsanis & Fisher, 1998; Stankiewicz et al, 2014). In most gene regulatory scenarios, CtBPs are considered as largely redundant proteins despite their diverging roles during mouse development (Hildebrand & Soriano, 2002; Stankiewicz et al, 2014). Nevertheless, in certain cases, such as in steatohepatitis, CTBP2 has been

attributed to unique gene regulatory functions (Sekiya et al, 2021). Although it is still unclear how this specificity is achieved, it might depend on the nuclear localization sequence, which is specifically present in CTBP2 but not CTBP1 (Verger et al, 2006).

Our genome-wide interactome study reveals CTBP2 interactions with members of the NF-κB (NFKB1, NFKBIZ, REL, RELA) and AP-1 (JUN, JUNB) families (Fig 3). This is supported by co-occupancy at shared chromatin locations for RELA and JUNB upon LPS stimulation (Fig 2). The interaction of CTBP2 with RELA may be regulated by RELA's nuclear localization upon LPS stimulation (Ernst et al, 2018; Bagaev et al, 2019). An inhibitory effect of CtBPs was previously observed for NF-κB luciferase reporters (Shen et al, 2017). Interestingly, we found that CTBP2 lost interactions with JUN and gained interactions with JUNB upon LPS stimulation, suggesting a co-regulator switch between those AP-1 subunits in inflammatory conditions. This might be attributable to a higher affinity for JUNB over JUN, as observed in NanoBRET assays (Fig 3D) or different kinetics of JUN and JUNB protein expression after LPS stimulation as observed previously in dendritic cells (Gomard et al, 2010). JUNB is considered as a modulator of macrophage inflammation and a repressor of cell cycle progression (Fontana et al, 2015), matching the transcriptional alterations observed after *Ctbp2* knockdown. Whereas JUNB can compensate for JUN during mouse development, it has been reported that JUNB partially antagonizes transcriptional effects of JUN on cytokine production in fibroblasts, causing the attenuated expression of specific inflammatory target genes (Szabowski et al, 2000; Passegué et al, 2002). This supports our observation that CTBP2 limits inflammatory gene expression of specific JUNB target genes, suggesting that CTBP2 fine-tunes JUNB-dependent inflammatory gene activation in macrophages. Furthermore, CTBP2 associates with transcriptional repressors to negatively regulate macrophage inflammation. We suggest that CTBP2 acts as a scaffold protein that brings co-repressors to specific genomic loci via its interaction with pro-inflammatory transcription factors such as NF-κB and JUNB. Among the identified co-repressors, the NuRD complex is able to repress *Ccl2* transcription and to antagonize SWI/SNF-mediated gene activation in macrophages (Pakala et al, 2010; Stabile et al, 2024 *Preprint*). In addition, the histone deacetylase 1 (HDAC1) has been reported to limit the expression of *IL12B* in HEK293T cells, potentially independent of the NuRD complex (Lu et al, 2005). Similarly, the histone demethylase KDM1A blunts inflammatory gene expression in non-macrophage cell lines (Janzer et al, 2012; Hanzu et al, 2013; Liu et al, 2024). In particular, KDM1A has been recognized as a repressor of inflammatory responses by restricting DNA binding of pro-inflammatory transcription factors such as RELA, which parallels our observation that JUNB DNA binding is negatively influenced by CtBPs (Hanzu et al, 2013; Kim et al, 2018; Wang et al, 2018; Liu et al, 2024). Altogether, our findings suggest a model in which CTBP2 bridges pro-inflammatory transcription factors and transcriptional co-repressors in activated macrophages to blunt inflammatory gene expression.

CtBPs are proposed as transcriptional integrators of the cellular energy state by metabolite-induced alterations in its oligomeric state (Zhang et al, 2002; Shen et al, 2017; Sekiya et al, 2021). Using oligomerization-defective mutants of CTBP2, we investigated whether different oligomeric states would differentially affect

transcription and CTBP2's protein–protein interactions. Importantly, higher order CTBP2 oligomers are required for its interaction with co-repressors such as KDM1A and for controlling inflammatory gene expression in macrophages (Fig 5F). However, the interaction with JUNB is preserved in CTBP2 monomers. Supporting our observations, CTBP monomers have been reported to lose the interaction with KDM1A in a mechanistic study in the human adenocarcinoma cell line HuTu80 (Ray et al, 2017). In contrast to wild type CTBP2 that is capable of forming dimers and tetramers required for the interaction with transcriptional co-repressors, monomeric CTBP2 appears to associate with RNA polymerase II, potentiating gene expression. This implies that CTBP2 may transition from a repressor into an activator of transcription by changing its oligomeric state. Nevertheless, whether metabolites such as Acyl-CoAs or NADH can alter CTBP2's oligomeric state in macrophages, rendering CTBP2 a sensor of cellular metabolism, remains to be investigated. It was recently shown that acetyl-CoA derivatives and fatty acids force CTBP2 monomerization in the liver of obese mice (Sekiya et al, 2021, 2023; Saito et al, 2023). This highlights the importance of the activator–repressor switch for metabolic disease, a condition associated with tissue inflammation (Johnson et al, 2012; Reddy et al, 2019; Kawai et al, 2021; Lee & Olefsky, 2021). Whether fatty acid induced CTBP2 monomerization also contributes to tissue inflammation by altering inflammatory gene expression in macrophages remains to be investigated. However, CTBP2 monomers specifically interacted with proteins involved in fatty acid metabolism in J774.1 cells (Fig 5D), supporting findings from Sekiya et al that binding of Acyl-CoAs to CTBP2 monomers inhibits dimer formation and indicating oligomerization-dependent, and potentially metabolite-dependent, complex associations (Sekiya et al, 2021).

Here, we focused our study on the prominent interaction between CTBP2 and JUNB but identified multiple other transcription factors such as NFKB1, CEBPB, or ETV3 in our interaction screen, as well as some of their footprints in CTBP2's cistrome. This indicates locus-specific functions and complex regulatory networks. In summary, our genome-wide profiling has identified CTBP2, but not CTBP1, as an important inflammatory co-regulator that assembles co-repressor complexes at sites of NF-κB and JUNB binding to balance pro-inflammatory gene expression programs in primary murine and human macrophages.

# Materials and Methods

### Isolation and culture of BMDMs

BMDMs were isolated as described previously (Barish et al, 2005). In brief, humerus, femur, and tibia were isolated from 6- to 12-wk-old C57BL6/N mice (RRID: MGI:7466658), disinfected with ethanol, and transferred to the cell culture hood. Using scissors, tweezers, and a syringe with a G27 needle, the bones were opened and the bone marrow was flushed into a 50-ml conical tube filled with cold RPMI 1640 (#R5886; Sigma-Aldrich). Cells were collected by centrifugation (300$g$, 5 min), the supernatant was aspirated, and cells were suspended in ACK lysis buffer (1 M NH$_4$Cl, 1 M KHCO$_3$, 0.5 M EDTA) to lyse red blood cells. Subsequently, cells were suspended in PBS,

carefully layered over Ficoll-Paque (#17144002; GE Healthcare), and subjected to density centrifugation (500$g$, 45 min, slow acceleration and deceleration). The top layer was aspirated, and the middle fraction was collected in a fresh conical tube filled with Differentiation Medium (30% L929 supernatant, 20% FBS, 1% penicillin/ streptomycin, 49% DMEM high glucose). Cells were pelleted by centrifugation, the supernatant was aspirated, and cells were suspended and plated in Differentiation Medium on 15-cm bacterial plates. After 3 d of incubation at 5% $CO_2$ and 37°C in a humidified incubator, half of the medium was replaced with fresh Differentiation Medium, and differentiation was continued until day 6. Cells were detached by incubation in Versene (#11518876; Thermo Fisher Scientific) and subjected to counting using Neubauer Hemocytometer. Finally, differentiated cells were seeded at 905,000 cells/ $cm^2$ in Macrophage-SFM medium (#12065074; Thermo Fisher Scientific) in tissue culture–treated dishes and incubated at 5% $CO_2$ and 37°C in a humidified incubator.

### Isolation and culture of human PBMCs

Experiments involving human volunteers were approved by the ethics committee of the Technical University of Munich (TUM). These studies were carried out in collaboration with the Core facility Human Studies at TUM. Two male and two female healthy volunteers between 25 and 40 provided 40 ml of blood each. White blood cells were isolated using density gradient centrifugation with Ficoll-Paque Plus. The centrifugation was performed at 2,000$g$ for 30 min without using the brake function. Subsequently, the white blood cell layer (ring) was carefully collected and transferred to a new tube. These cells were then washed twice with a generous volume of D-PBS and centrifuged at 500$g$ for 10 min. Cell count was determined using Neubauer Hemocytometer. The cells were then seeded at twice the desired experimental density in Macrophage-SFM medium (#12065074; Thermo Fisher Scientific), which was supplemented with 40 ng/ml human M-CSF (#300-25; PeproTech). The cells were cultured for 3 d at 37°C and 5% $CO_2$.

### siRNA knockdown in PBMCs and BMDM

siRNAs (Dharmacon, Table S1) were solubilized in nuclease-free water at 20 $\mu$M and further diluted in Opti-MEM (#31985047; 1.25 + 50 $\mu$l; Thermo Fisher Scientific) to a final concentration of 50 nM. RNAiMAX (#13778075; Thermo Fisher Scientific) was prediluted in Opti-MEM (1 + 50 $\mu$l), and equal volumes of siRNA dilution and RNAiMAX dilution were mixed thoroughly by pipetting. For knockdowns in BMDMs, 100 $\mu$l of this mix was added to each well of a 24-well plate and incubated at RT for 20 min to allow for complex maturation. In the meantime, cells were diluted to 500,000 cells/ml and 400 $\mu$l cell suspension was added to the complex and mixed by rocking the plate back and forth. The next day, medium was replaced by fresh Macrophage-SFM and wells were incubated for a total of 72 h before assessing knockdown efficacy by RT–qPCR or Western blot. In case of stimulation experiments, cells were stimulated 68 h after knockdown. For PBMCs, mature RNAiMAX/ siRNA complexes were added directly to cells at the end of differentiation. To this end, growth medium was replaced with 400 $\mu$l RPMI 1640 supplemented with 10% FBS and 50 mM Gln and 120 $\mu$l

complex was added directly to the cells. Transfected cells were incubated for 72 h before assessment of knockdown effects. 6 h after transfection, medium was replaced with normal growth medium.

### Stimulation of cells

Cells were seeded and incubated overnight at normal growth conditions. Then, vehicle (PBS) or LPS (#LPS25, 100 ng/ml; Sigma-Aldrich) was added, and cells were incubated for 3 h for ChIP experiments and protein isolation or for 4 h for assessment of transcriptional effects using RNA.

### Western blot

Cells were lysed in RIPA buffer (150 mM NaCl, 50 mM Tris [pH 7.4], 1% Nonidet P-40, 0.5% sodium deoxycholate, 0.1% sodium dodecyl sulfate [SDS]). 6X Laemmli buffer (375 mM Tris–HCl [pH 6.8], 6% SDS, 4.8% glycerol, 9% 2-mercaptoethanol, and 0.03% bromophenol blue) was added to a final concentration of 1X, and protein lysates were boiled for 10 min at 95°C. Proteins were separated by size using an 8% polyacrylamide gel in running buffer (25 mM Tris, 192 mM glycine, 0.1% SDS) and transferred to a PVDF membrane in transfer buffer (25 mM Tris, 192 mM glycine, 20% ethanol) using semi-dry transfer (25 V, 150 min, Trans-Blot SD; Bio-Rad;). Membranes were blocked for 1 h in 5% BSA in TBS-T (150 mM NaCl, 10 mM Tris, 0.1% Tween-20) and incubated with primary antibody (Table S2) overnight at 4°C. The primary antibody was removed, and membranes were washed thrice with TBS-T before incubation with HRP-coupled secondary antibodies (Table S2) for 1 h at ~20°C. Membranes were washed again thrice with TBS-T before HRP substrate was added. Chemiluminescence was measured using Sapphire Azure Biomolecular Imager (Azure Biosystems).

### RNA extraction and reverse transcription

RNA was isolated using the ReliaPrep RNA miniprep system (#Z6012; Promega) following the manufacturer's instructions. In brief, cells were lysed in BL+TG buffer and lysates stored at −70°C. For isolation of RNA, lysates were mixed with 2-propanol, bound to a silica membrane, and washed, and DNA was digested by DNase I. After additional washes, RNA was eluted in nuclease-free water. RNA concentration was measured using a NanoPhotometer (Implen), and 100–1,000 ng of RNA was reverse-transcribed using the Reverse Transcription System (#A5001; Promega) with random hexamers following the manufacturer's instructions. A control without reverse transcriptase was included to assess potential DNA contaminations during qPCR.

### qPCR

Small aliquots of cDNA were taken from each sample, pooled, and diluted 1:3 for generation of standard S1. Standards S2-S5 were prepared by serial dilution of S1 (1:5 each). The remaining cDNA was diluted 1:10. qPCRs including a water control were performed using gene-specific primers (Table S3) and the qPCR system (#A6002; Promega) in CFX384 Real-Time PCR Detection System (Bio-Rad)

following the manufacturer's instructions. Transcript homogeneity was assessed by melting point analysis, and data were analyzed in R 4.3.0 (R Core Team, 2018) using homemade scripts and the dCT method.

## RNA-seq

RNA quality was determined on Agilent 2100 Bioanalyzer with the RNA 6000 Nano kit (#5067-1511; Agilent), following the manufacturer's instructions. Library preparation and rRNA depletion were conducted using the TruSeq unstranded mRNA Library Prep kit (Illumina), starting with 500 ng of RNA for each biological triplicate. The samples were sequenced on the Illumina NovaSeq 6000.

### Data analysis
NGS data quality was assessed with FastQC (RRID: SCR 014583, http://www.bioinformatics.babraham.ac.uk/projects/fastqc/).

For RNA-seq, the gene-level quantification was performed with Salmon version 1.9.0 (RRID: SCR_017036 [Patro et al, 2017]). Settings were as follows: -libType A, -gcBias, -biasSpeedSamp 5 using the mm39 (M28, GRCm38, mm39) reference transcriptome provided by Gencode (Frankish et al, 2021). Gene count normalization and differential expression analysis were performed with DESeq2 version 1.44.0 (RRID: SCR_015687 [Love et al, 2014]) after import of gene-level estimates with "tximport" version 1.32.0 (RRID: SCR_016752 [Soneson et al, 2015]) in R (RRID: SCR_001905, R version 4.4.1).

For gene annotation, Ensembl gene IDs were mapped to MGI symbols using the Bioconductor package "AnnotationHub" version 3.12.0 (RRID: SCR_024227), and genome information was provided by Ensembl (GRCm39 release 105). Genes with at least 1 read count, baseMean > 100, fold change of 1.4, and $P < 0.05$ were called significantly changed. Plots were generated with "ggplot2" version 3.4.4 (RRID: SCR_014601 [Wickham, 2016]), and GO enrichment was performed with "clusterProfiler" version 4.10.0 (RRID: SCR 016884 [Yu et al, 2012]).

## Chromatin immunoprecipitation (ChIP)–qPCR

$2 \times 10^7$ cells were fixed with 2 mM DSG for 30 min at RT followed by 10 min with 1% formaldehyde. Cross-linking was quenched by the addition of 85 mM glycine and 5-min incubation. Cross-linked cells were washed twice with ice-cold PBS, scraped and collected in microcentrifugation tubes, and pelleted at $1,000g$ for 5 min at 4°C. The supernatant was aspirated, and cell pellets were frozen at −70°C. Cells were thawed by the addition of 1 ml Fast IP buffer (167.5 mM NaCl, 5 mM EDTA, 50 mM Tris, pH 7.5, 1% Triton X-100, 0.5% Nonidet P-40, proteinase inhibitor) and suspended by pipetting. The cell suspension was incubated on ice for 10 min to destabilize cell membranes and pulled through an insulin syringe during that time to lyse cells. Cells were pelleted, and the previous step was repeated once. Then, the pellet was suspended in 1 ml shearing buffer (10 mM EDTA, 50 mM Tris, pH 8.0, 1% SDS) to lyse nuclei and free cross-linked chromatin. The solution was divided into two 500 µl aliquots and transferred into 1.5 ml Diagenode TPX tubes. Samples were then subjected to 18 cycles of sonication (30 s on/30 s off) at 4°C using the Bioruptor Plus (Diagenode) at "high" to generate DNA fragments of 500–1,000 bp. Chromatin was cleared

from precipitates by centrifugation (10 min, $14,000g$, 4°C) and diluted 1:10 in dilution buffer (168 mM NaCl, 1.2 mM EDTA, 16.7 mM Tris, pH 8.0, 0.01% SDS, 1.1% Triton X-100). Then, 1 ml chromatin dilution was incubated with target-specific antibody (anti-CTBP2, anti-JUNB, anti-JUN, anti-RELA, anti-NFKB1, IgG; Table S2) at 4°C overnight. The next day, lysates were cleared from precipitates by centrifugation ($14,000g$, 10 min, 4°C), and the top 90% were carefully transferred to a fresh 1.5-ml low DNA-binding microcentrifugation tube prefilled with 10 µl Dynabeads (#11204D; Thermo Fisher Scientific) in dilution buffer. Tubes were rotated at 4°C for 5 h to immobilize protein–DNA–antibody complexes at the bead surface. Subsequently, beads were placed on a magnet and the supernatant was removed by aspiration. Beads were washed six times with ice-cold dilution buffer and one time with TE buffer (#T9285; Sigma-Aldrich) before elution of targeted protein–DNA complexes in bead elution buffer (100 mM $NaHCO_3$, 1% SDS). Cross-linking was reversed by incubation with 195 mM NaCl at 65°C overnight. RNA was removed by the addition of RNase A (50 ng/µl) and incubation for 30 min at 37°C, and proteins were digested by the addition of proteinase K (45 ng/µl) and incubation at 56°C for 1 h before DNA was cleaned using the ChIP and DNA concentrator (#D5205; Zymogen) following the manufacturer's instructions. ChIP DNA was eluted in nuclease-free water and immediately used for qPCR. Cq-values for ChIP samples and respective inputs were assessed via qPCR in technical triplicates. The mean of technical triplicates was normalized to the respective input in percent. The mean percent input of three independent experiments (with two technical replicates each) was plotted as a bar with the SD as an error bar and single dots as individual replicates.

## ChIP-seq

ChIP-seq was performed as ChIP-qPCR with the following modifications: $4 \times 10^7$ cells were used for a single immunoprecipitation, and after sonication, 16 ml chromatin dilution was incubated with 10 µl CTBP2-specific antibody (#61261; Active Motif) at 4°C overnight. After clean-up, ChIP DNA was eluted in nuclease-free water and frozen at −20°C.

### Library preparation
The DNA was quantified via Qubit, and the enrichment was validated by qPCR. Libraries were performed with the NEB Next Ultra II DNA kit according to the manufacturer's instructions and sequenced on an Illumina NovaSeq 6000 machine at 100-bp paired ends.

### Data analysis
NGS data quality was assessed with FastQC (RRID: SCR 014583, http://www.bioinformatics.babraham.ac.uk/projects/fastqc/).

ChIP-seq paired-end reads were mapped to the murine reference genome mm39 (Ensembl GRCm39.p6 [Cunningham et al, 2019]) with BWA-MEM version 0.7.17 (RRID: SCR 010910 [Li, 2013 Preprint]), and PCR duplicates were removed using Picard Tools version 3.2.0 (RRID: SCR 006525, http://picard.sourceforge.net/). For visualization, bam files were filtered for properly paired and mapped reads, and multimappers were removed with SAMtools version 1.9 (RRID: SCR 002105 [Li et al, 2009]). Alignments were converted to bigwig files, merging 10 bp per bin using "bamCoverage" from the

deepTools package version 3.5.1 (RRID: SCR 016366 [Ramírez et al, 2016]). Tracks were visualized with the UCSC Genome Browser (RRID: SCR 005780 [Perez et al, 2025]). Peaks were called with MACS version 3.0.0a5 in BAMPE mode and an FDR cutoff of 0.05. ChIP-seq peaks were called overmatched input controls. Blacklisted regions (lifted from http://mitra.stanford.edu/kundaje/akundaje/release/blacklists/mm10-mouse/mm10.blacklist.bed.gz to mm39 using UCSC liftOver [Perez et al, 2025]) were removed from analyses. Peaks were annotated to the closest gene expressed in macrophages in any of our conditions with the "ChIPpeakAnno" package version 3.40.0 (RRID: SCR 012828 [Zhu et al, 2010]) in R version 4.4.1 (RRID: SCR 014601 [R Core Team, 2018]) and annotation data from the mouse Ensembl genome GRCm39 release 112 (mm39 [Cunningham et al, 2019]). Genes were called expressed when passing a mean expression value of the 25th percentile. Motif enrichment was performed on peaks trimmed to 100 bp around the peak center with HOMER (RRID: SCR 010881 [Heinz et al, 2010]). Differential binding analysis of LPS-treated versus vehicle-treated CTBP2 cistromes was performed using DiffBind version 3.16 (RRID: SCR 012918 [Ross-Innes et al, 2012]).

### ChIP-MS

ChIP-MS was performed as ChIP-qPCR with the following modifications: $6 \times 10^7$ cells were used for immunoprecipitation using anti-CTBP2 (#61261; Active Motif) and an IgG control antibody, and after sonication, 16 ml chromatin dilution was incubated with 10 $\mu$l antibody at 4°C overnight. After immobilization of protein complexes on Dynabeads, these were washed thrice with low salt buffer (140 mM NaCl, 50 mM Hepes, pH 7.5, 1% Triton X-100) and once with high salt buffer (500 mM NaCl, 50 mM Hepes, pH 7.5, 1% Triton X-100). To remove detergents, beads were washed twice with TBS (150 mM NaCl, 50 mM Tris, pH 7.4). The supernatant was removed completely, and beads were immediately frozen on dry ice. Samples were then shipped to the Proteomics Research Infrastructure at the University of Copenhagen and subjected to their pipeline to remove remaining nucleic acids and generate peptides for mass spectrometry.

Beads were incubated for 30 min with elution buffer 1 (2 M urea, 50 mM Tris–HCl, pH 7.5, 2 mM DTT, 20 $\mu$g/ml trypsin) followed by a second elution with elution buffer 2 (2 M urea, 5 mM Tris–HCl, pH 7.5, 10 mM chloroacetamide) for 5 min. Both eluates were combined and further incubated at RT overnight. Tryptic peptide mixtures were acidified to 1% TFA and loaded onto Evotips (Evosep) for LC-MS analysis.

Peptides were injected into a Bruker timsTOF Pro2 mass via a CaptiveSpray source with a 20 $\mu$m emitter. Data acquisition was performed in PASEF mode with a mass range of 100–1,700 m/z and a TIMS mobility range of 0.6–1.6 1/K0. Three Agilent ESI-L Tuning Mix ions were used to calibrate the ion mobility: 622.0289, 922.0097, and 1,221.9906. The TIMS ramp and accumulation times were set to 100 ms each, and 10 PASEF ramps were recorded, resulting in a total cycle time of 1.17 s. The MS/MS target intensity was set to 20,000, and the intensity threshold was set to 2,500. An exclusion list of 0.4 min was activated for precursors within 0.015 m/z and 0.015 V cm$^{-2}$ width.

### Data analysis

All statistical analysis of protein expression intensity data was done with in-house Python code from the Clinical Knowledge Graph's automated analysis pipeline (Santos et al, 2022). Potential contaminants, as well as proteins identified by matches to the decoy reverse database or only by modified sites, were removed. Intensities were $\log_2$-transformed, and proteins with fewer than two valid values in at least one group were excluded. Missing values were imputed using the MinProb approach (width = 0.2 and shift = 1.8), as described in Lazar et al (2016). Differentially expressed features were identified by statistical unpaired $t$ tests, and Benjamini–Hochberg correction for multiple hypothesis testing with false discovery rate (FDR) threshold 0.05 and fold change of 4. Gene ontology enrichment analysis was performed using "UniprotR" version 2.4.0 (RRID: SCR 023483 [Soudy et al, 2020]).

### Culture of cell lines

J774.1 (RRID: CVCL_4770) cells were cultured in DMEM high glucose (#D6429; Sigma-Aldrich) supplemented with 10% FBS (#S0615; Sigma-Aldrich) and 1% penicillin/streptomycin (#P4333; Sigma-Aldrich) at 5% $CO_2$ and 37°C in a humidified incubator on tissue culture–treated dishes. At ~70–80% confluence, cells were passaged by scraping and suspending in fresh medium prewarmed to 37°C.

### Generation of *Ctbp1/2* knockouts in J774.1 cells

To assemble a functional sgRNA, 2 $\mu$l Alt-R CRISPR/Cas9 tracrRNA labeled with ATTO 550 and 2 $\mu$l crRNA (IDT, 100 $\mu$M) were mixed in 20 $\mu$l duplex buffer, heated to 95°C for 5 min, and slowly cooled back to RT within 2 h. 0.7 $\mu$l annealed oligonucleotides (50 $\mu$M) were mixed with 0.5 $\mu$l Alt-R Cas9 (IDT) in 1.8 $\mu$l PBS and incubated for 20 min at RT to assemble RNP complexes. 500,000 cells of the cell line J774.1 were washed with PBS and suspended in 12 $\mu$l electroporation buffer R (Thermo Fisher Scientific). 4 $\mu$l of electroporation enhancer (15 $\mu$M IDT), 8 $\mu$l of electroporation buffer R, and 12 $\mu$l cell suspension were added to the assembled RNP complexes. The solution was mixed with a 10-$\mu$l electroporation tip. The filled tip was placed in the Neon electroporation device (Thermo Fisher Scientific) in 5 ml buffer E and subjected to 3 × 1,400 V pulses for 10 ms each. Then, cells were seeded in six-well plates prefilled with full growth medium. The next day, cells were suspended in FACS buffer (5 mM EDTA, 2% FBS, PBS), pipetted through a 70-$\mu$m cell strainer to obtain a single-cell suspension, and subjected to fluorescence-activated cell sorting (FACS, BD Aria II). To enrich successfully transfected cells, ATTO 550–positive cells were sorted in a container filled with growth medium. Cells were diluted to 20 cells/ml, and 100 $\mu$l aliquots were seeded on 96-well plates for single-cell outgrowth. During this time, cells were monitored closely to ensure monoclonal origin. After reaching confluence, clonal colonies were propagated and subjected to genotyping by PCR (primers in Table S3). Selected clones were confirmed as either WT or knockout by sequencing of PCR products and Western blotting.

## Immunofluorescence

Cells were fixed in 4% formaldehyde for 15 min, washed with PBS, and stored for up to 1 wk at 4°C. Then, cells were blocked in blocking buffer (1% BSA, 0.1% Triton X-100, 0.05% Tween-20, PBS) for 1 h and incubated with primary antibody (Table S2) diluted in blocking buffer overnight at 4°C. Cells were washed thrice with PBS and incubated in secondary antibody (Table S2) diluted in blocking buffer for 1 h at RT. Nuclei were counterstained with DAPI (500 ng/ml) and washed thrice with PBS. Stained cells were stored at 4°C in the dark for up to 1 wk before imaging. Imaging was performed using the FV3000 confocal laser-scanning microscope (Olympus) with diode lasers 405 and 488 nm and Olympus four-channel TruSpectral detection system. Images were loaded and compiled using Fiji (SCR_002285) and the "Quick Figures" plug-in (Mazo, 2021).

## Generation of plasmids

*CTBP2* coding sequence (CDS) was cloned from human cDNA into pBiFC-VN155(I152L) (a gift from Chang-Deng Hu [RRID: Addgene_27097] [Kodama & Hu, 2010]), and mutants were generated by site-directed mutagenesis (SDM, all primers listed in Table S3). Then, *CTBP2* CDS was subcloned into PiggyBac, pcDNA3.1, pHTC Halo-tag, and pNLF1_C (Promega) plasmids. *Rela*, *Jun*, and *Junb* CDS were cloned from murine cDNA using the primers in Table S3 into pNLF1-N and pcDNA3.1 for NanoBRET and luciferase assays, respectively. Luciferase reporter constructs were cloned by amplification of respective regulatory sequences from murine gDNA via nested PCR and ligation into pGl3.basic. Mutated AP-1 and NF-κB binding sites were generated by SDM. Integrity of every inserted sequence was verified by sequencing (Eurofins). All plasmids are listed in Table S4.

## Transfection of cells using FuGENE

Cells were seeded at $8 \times 10^5$ cells/well in six-well plates in 2 ml full growth medium and maintained under normal growth conditions for 6 h. Then, 2 µg of plasmid (Table S4) was diluted in 100 µl Opti-MEM and 6 µl FuGENE (#E2311; Promega) was added and immediately mixed by pipetting. Complexes were allowed to form for 10 min at RT, and transfection mix was added dropwise to cells in six-well plates. Cells were placed back into the incubator and maintained at normal growth conditions until performing downstream experiments (typically 18–24 h). For smaller well sizes, transfection reactions were scaled down in proportion to the growth area.

## Generation of rescue cells

J774.1 *Ctbp* dKO cells or *Ctbp2* KO cells were seeded in 96-well plates and transfected with a plasmid coding for PiggyBac transposase and PiggyBac plasmid encoding puromycin-N-acetyltransferase and CTBP2 (or a mutant thereof) and incubated overnight at normal growth conditions. The next day, medium was replaced with fresh medium containing 10 µg/ml Puromycin (#P9620-10ML; Sigma-Aldrich) to select for successfully transfected cells. Selection medium was changed every 2nd d within the 1st wk to remove dead cells and

maintain selection pressure. Upon 70% confluence, cells were passaged to bigger plates and analyzed in regard to genotype (PCR + sequencing) and protein expression (Western blot). Selection pressure was maintained throughout culture of rescued cell lines but terminated when seeding for an experiment.

## NanoBRET assay

J774.1 *Ctbp* dKO cells were transfected with pHTC-CTBP2 in combination with either pNLF1-N-JUNB, pNLF1-N-RELA, pNLF1-N-JUN, or pNLF1-C-CTBP2 as described above. The next day, cells were scraped from six-well plates, washed, counted, and resuspended at 800,000 cells/ml in Opti-MEM supplemented with 4% FBS. 50 µl cell suspension was further diluted with 150 µl Opti-MEM supplemented with 4% FBS and 0.2 µl DMSO, or 0.2 µl Halo-ligand (#N1661; Promega) was added together with LPS (100 ng/ml). Cell suspensions were mixed, and 40 µl aliquots were plated in wells of a 384-well white flat-bottom plate (Corning). Cells were incubated for 6 h at normal growth conditions before plates were measured for total luminescence and luminescence through a 620-nm filter in the absence of NanoBRET-Glo Substrate (#N1661; Promega). Then, the substrate was added to wells (1:500), plates were rocked back and forth to mix, and the plate was measured again. BRET was assessed by dividing luminescence at 620 nm by total luminescence * 1,000 for each well in order to obtain milliBRET units (mBU), and then, values from wells without ligand were subtracted to correct for bleed-through effects.

## Luciferase reporter assay

J774.1 *Ctbp* dKO cells were cotransfected with pRenilla_CMV, pGl3.basic_Il1a_promoter, or respective mutants for NF-κB or AP-1 binding sites and either pcDNA3.1 or pcDNA3.1_CTBP2, as described above. The next day, cells were plated on wells of a 96-well plate white flat-bottom half-area and stimulated with LPS (100 ng/ml). After a total of 48 h after transfection, luminescence of Firefly luciferase and Renilla luciferase was measured using the Dual-Glo luciferase assay system (#E2940; Promega) and Tecan M-200 following the manufacturer's instructions. Relative luciferase activity was calculated by dividing Firefly luciferase signal (AU) by Renilla luciferase signal (AU) for each well.

## Software

R (4.3.0)
  Fiji (2.16.0, incl. ImageJ [1.54f] and Quick Figures plug-in).

# Data Availability

The RNA, ChIP, and MS data produced in this study are available in the following databases: RNA-seq data: Gene Expression Omnibus GSE287789; ChIP-seq data: Gene Expression Omnibus GSE287719; protein interaction AP-MS data: the mass spectrometry proteomics data have been deposited to the ProteomeXchange Consortium via the PRIDE (Perez-Riverol et al, 2025) partner repository with the dataset identifiers PXD063850 and PXD063883.

# Supplementary Information

# Acknowledgements

We are very grateful to all the volunteers and blood donors who participated in this study. Moreover, we sincerely thank R Scheundel, K Petzold, R Müller, M Schweiger, T Horn, I Guderian, E Eichinger, S Sharma, J Ukamaka, M Fink, S Regn, and J Brehme for their contributions to this study, and B Spanier for helpful discussions. NGS data were generated at the DRESDEN-concept Genome Center, supported by the DFG Research Infrastructure Program (Project 407482635), at the Next Generation Sequencing Competence Network (NGS-CN; Project 423957469), and at the HMGU Genomics Core Facility of Helmholtz Munich (Dr. Inti Alberto de la Rosa Velazquez). Mass spectrometry–based proteomic analyses were performed by the Proteomics Research Infrastructure (PRI) at the University of Copenhagen (UCPH), supported by the Novo Nordisk Foundation (NNF) (grant agreement number NNF19SA0059305). Confocal imaging was performed at the Center for Advanced Light Microscopy (CALM) of the TUM School of Life Sciences. This project received funding from the German Research Foundation DFG (SFB 1064 Chromatin Dynamics [Project-ID 213249687]), the DFG project 497680553 to F Greulich, the DFG projects 490946138 and 314061271 to NH Uhlenhaut and from TUM Global Incentive Funds. P Singh is supported by the International Helmholtz-Edinburgh Research School for Epigenetics with funding to F Greulich.

## Author Contributions

BA Strickland: conceptualization, resources, data curation, software, formal analysis, validation, investigation, visualization, methodology, project administration, and writing—original draft, review, and editing.
A Babl: investigation, visualization, and methodology.
L Wolff: investigation, visualization, methodology, and writing—review and editing.
P Singh: visualization.
ME Friano: investigation and methodology.
F Greulich: conceptualization, resources, data curation, software, formal analysis, supervision, funding acquisition, visualization, methodology, project administration, and writing—original draft, review, and editing.
NH Uhlenhaut: conceptualization, supervision, funding acquisition, and writing—original draft, review, and editing.

## Conflict of Interest Statement

The authors declare that they have no conflict of interest.

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
