## [Reviewer comments · Life Science Alliance]

Life Science Alliance

C-terminal binding protein 2 interacts with JUNB to control macrophage inflammation

Benjamin Strickland, Antonia Babl, Lena Wolff, Priya Singh, Marika Friano, Franziska Greulich, and Nina Uhlenhaut
DOI: <https://doi.org/10.26508/lsa.202503263>

Corresponding author(s): Nina Uhlenhaut, Technical University of Munich and Franziska Greulich, TUM/School of Life Sciences Weihenstephan

Review Timeline:

Submission Date:	2025-02-10
Editorial Decision:	2025-03-13
Revision Received:	2025-05-19
Editorial Decision:	2025-05-20
Revision Received:	2025-05-23
Accepted:	2025-05-26

Scientific Editor: Tim Fessenden

Transaction Report:

March 13, 2025

Re: Life Science Alliance manuscript #LSA-2025-03263-T

Prof. Nina Henriette Uhlenhaut
TUM School of Life Sciences Weihenstephan
Metabolic Programming
Gregor-Mendel-Str. 2
Freising 85354
Germany

Dear Dr. Uhlenhaut,

Thank you for submitting your manuscript entitled "C-terminal binding protein 2 interacts with JUNB to repress macrophage inflammation" to Life Science Alliance. The manuscript was assessed by expert reviewers, whose comments are appended to this letter. We invite you to submit a revised manuscript addressing the Reviewer comments.

Thank you for this interesting contribution to Life Science Alliance. We are looking forward to receiving your revised manuscript.

Sincerely,

B. MANUSCRIPT ORGANIZATION AND FORMATTING:

Reviewer #1 (Comments to the Authors (Required)):

This study build on previous work from the same lab identifying CTBP2 function in transcriptional regulation of inflammation by glucocorticoids(Greulich et al, 2021). The response to glucocorticoids is complex. Many of the actions in macrophages are due to transcriptional induction of feedback regulators such as DUSP1. In macrophages GR binding sites are associated with binding sites for the macrophage-specific transcription factor, PU.1 and glucocorticoid action is associated with sustained chromatin decompaction (PMID: 29241532. PMID: 26663721). The current study is focused more on negative regulation of the response to lipopolysaccharide (LPS). The authors should place this work more in the context of known feedback regulation of this response. Detailed analysis of the LPS response in human macrophages revealed a sequential cascade of transient induction and repression over a very extended time course (PMID: 28263993). It is likely that CTBP2 contributes to that feedback regulatory cascade. The authors might also refer to the known roles of HDACs and the significant literature on pharmacological inhibitors in inflammation. (PMID: 21570914;PMID: 26900475). The putative targets of Ctbp2 repression overlap with the selective impact of HDAC inhibitors.

Overall, the work is technically competent and convincing providing multiple lines of evidence to support the proposition that oligomeric CTBP2 represses a subset of LPS-inducible genes in macrophages.

It appears that results and discussion have been combined. I think that the story would be easier to follow if there were separate results and discussion sections and discussion was strictly removed from the results.

One minor technical question. In the methods the bone marrow-derived macrophages are generated in a rather unconventional medium (to me at least) involving high concentrations of L929 conditioned medium and serum. The cells are then transferred to what is called macrophage serum free medium. This is not described in any detail and the product description on Thermo Fisher is rather nebulous. Following back through two citations, it appears that the medium contains 10ng/ml M-CSF as well as various additives. The authors should cite the original papers and clarify the composition.

Two references are incomplete and 2019 paper on BioRxiv (Li et al) is published PMID: 32174788

Reviewer #2 (Comments to the Authors (Required)):

Strickland et al. provide a comprehensive study of the transcriptional corepressor role of the nuclear protein CTBP2 in macrophages. Strengths of the work include the use of multiple mechanistic levels of analysis (RNAseq, ChIPseq, ChIP-MS, nanoBRET) in cells depleted of CTBP2 or the related CTBP1 by RNAi or sgRNAs and reconstituted with wild-type or dimerisation/oligomerisation-deficient point mutants of CTBP2. All these data are finally integrated into a model showing that JUNB may act as a recruitment factor for CTBP2 to specific genomic loci in enhancers or promoters of prototypical pro-inflammatory LPS-induced genes such as Il1a or Ccl22, in order to repress gene expression in its multimerised form. In conclusion, this study provides compelling evidence for a bipartite recruitment mechanism of CTBP2, whereby TFs such as JUNB recruit the protein and its oligomerisation provides a platform for further recruitment of transcriptional repressor complexes. These findings represent interesting new information on negative gene regulatory mechanisms in the innate immune response. The manuscript is well written and the data are consistent with the author's conclusions.

Specific points:

Fig. 3D. Why was this experiment performed in murine fibroblasts? The author should add data from untreated cells to show whether the interaction of JunB with Ctbp2 is regulated by LPS. Immunoblots should be provided to show equal levels of fusion protein expression under all conditions.

Referee Cross-Comments:

I agree with most of the other reviewer's comments and suggest that it should be sufficient to address them by modifying the manuscript text and some of the data presentations accordingly.

I do not believe that these comments justify extensive additional experiments in this already large study.

Reviewer #3 (Comments to the Authors (Required)):

Strickland et al. study the roles of the the coregulators CTBP1/2 in mouse and human macrophage activation, find CTBP2 to be involved in inflammatory gene regulation but not CTBP1. CTBP2 binds to CREs along with NF- κ B and AP-1 and requires dimerization to limit transcription activation by TLR4 activation, likely via interactions with other repressor proteins. Finally, the authors show that a monomeric CTBP2 mutant continues to interact with JUNB and RNA Pol II and leads to further increased transcription activation by LPS.

Overall, the study provides a plethora of data to assess the contributions of CTBP1 and CTBP2 to gene activation by LPS in macrophages that will be interesting for the field. The manuscript would benefit from adjusting some of the language describing the findings and conclusions to match the data, additional ways of presenting the data, and expanding the discussion of prior data.

Major concerns:

1. Inaccurate nomenclature is confusing and sometimes misleading.

- a) The authors show that in the absence of CTBP2, LPS-mediated induction of transcription of many genes is augmented. However, given that these genes are also highly transcriptionally activated in the presence of CTBP2 (10-1000-fold (e.g. PMID: 29403501)), it would be more accurate to call CTBP2 a limiter of inflammatory gene activation throughout the manuscript, rather than "a repressor of inflammatory gene expression".
- b) The subtitle "CTBP2 binds JUNB and RELA motifs close to inflammatory genes in macrophages" inadvertently suggests that CTBP2 recognizes and directly binds DNA, which is not the case.
- c) While KDM1A has been shown to demethylate some transcription factors (e.g. p53, HIF-1 α), I am not aware that KDM1A was described to demethylate RELA in any of the papers cited by the authors: "KDM1A has been recognized as a repressor of inflammatory responses by demethylation of pro-inflammatory transcription factors such as RELA (Hanzu et al., 2013; Liu et al., 2024; Wang et al, 2018)"

2. Incorrect or overstated conclusions.

- a) The authors' conclusion that JUNB recruits CTBP2 to repress or down-modulate transcription of LPS-stimulated genes (IFN1 α , IL12b) in macrophages contradicts previous findings that JunB was required for high-level LPS-driven transcriptional activation of proinflammatory cytokines such as IL1b, IL12b and TNF (Fontana, JI 2015). The authors cite Fontana et al. for mentioning the role of JUNB in proliferation but do not discuss or reconcile these discrepancies, especially with regard to their interpretation that JUNB exerts "anti-inflammatory actions". This disagreement with the literature would need to be addressed in the discussion.
- b) Text incongruent with data: Fig. 1E shows CCL22 not significantly increased with CTBP2 KD, and for sure not "hyper-activated", and none of the changes in IL12B mRNA with CTBP2 KD in Fig. S1E were statistically significant ($p = 0.1$). However, the manuscript states: "Indeed, knockdown of CTBP2 but not CTBP1 led to hyper-activation of CCL22 and IL12B following LPS treatment (Fig. 1E, Fig. S1E)." - this conclusion is not adequate and should be corrected to reflect the presented data.
- c) While the authors show that JUNB and CTBP2 interact, they do not show that e.g. CTBP2 fails to be recruited in the absence of JUNB, e.g. following JUNB KO. In the absence of this data, conclusions along the lines that "CTBP2 is recruited ... by JUNB" should probably be weakened to "Our data strongly indicates that CTBP2 is recruited ... by JUNB".
- d) The manuscript tries to arrive at significant conclusions based on often very small quantitative differences of uncertain significance. For example, the conclusion that CTBP2 knockout increases JUNB binding is based on very small differences in enrichment measured by ChIP-qPCR (Fig. 4E) that are interpreted to reflect significant changes in JUNB binding to different genomic locations. However, the description of the experimental methodology and the significance calculation need to be improved. Specifically, the %input measurement used for t-test is a derived measurement that incorporates both a IP signal and an input signal. These independent variables are encumbered by individual measurement errors that contribute to the overall error in the derived %input measurement, which therefore is larger than the individual measurement errors of each value. However, whether and how the errors were propagated is not described. Further, it is not clear whether the values shown are from independent experiments with different KO cell populations (i.e. biological replicates) or technical ChIP replicates with the same cell population, or technical PCR replicates from the same ChIP. Together with the minute reported differences in ChIP-qPCR signal between different cell clones/populations (equivalent to less than 0.5 cycles in qPCR), this puts into question the significance of the small ChIP-qPCR differences and weakens the conclusion.
- e) The conclusion: "In contrast to the CTBP2 cistrome in murine livers (Sekiya et al., 2021), we find that in macrophages CTBP2's cistrome is enriched for motifs of inflammatory transcription factors indicating that CTBP2 binding is highly tissue-specific.", is based on misinterpretation of the data and should be corrected. For example, the fact that the subset of LPS-inducible CTBP2 binding locations are enriched for motifs of LPS-activated TFs NF- κ B and AP-1, while CTBP2 binding locations in unstimulated liver cells are not, does not mean that CTBP2 binding is tissue-specific. To make that statement, the authors would need to compare motif enrichment in either LPS-inducible macrophage-specific CTBP2 peaks to the inducible peaks in

liver cells similarly stimulated with LPS, or compare the motifs associated with the basal CTBP2 binding patterns in macrophages (~20k sites) and liver cells.

Minor concerns and suggestions:

The ChIP-MS protocol is not provided, and the corresponding citation in the Methods section is missing ("Samples were then shipped to the proteomics research infrastructure Copenhagen and subjected to their pipeline to remove remaining nucleic acids and generate peptides for mass spectrometry as described previously (REF)."). Due to the missing protocol, it is not clear how/whether the samples were DNase-treated to only measure direct protein-protein interactions.

The DNA binding motifs for c-Jun and JunB are the same, therefore the statement that motif differences correlate with differences in binding is not valid. ("Contrarily, we did not observe binding of NFKB1 or JUN at any investigated site, which is in line with the motif enrichment results from our ChIP-seq experiments (Fig. 2D).")

From the way the RNA-seq data in Fig. 1/S1 is presented, as a z-scored heat map of relative expression values of the most significantly changing genes with and without LPS treatment, or a volcano plot of differences in LPS-induced relative expression values, it is difficult to verify the conclusion that CTBPs repress transcription, as it is impossible to gauge the absolute gene expression differences induced by LPS in the presence or absence of CTBP1/2. A supplemental figure showing e.g. a MA-style plots of averaged vehicle vs. LPS treatment RNA-seq data for wt, CTBP1 and CTBP2 KD would enable the reader to get a better picture of the magnitude differences in absolute transcript levels caused by disrupting CTBP1/2.

The authors could use NanoBRET to show that CTBP2 does not interact with Jun.

Jun and NFKB1 ChIP-qPCRs should include positive controls.

Typos:

"RAW264.7", not "Raw246.7"

"GSE38379", not "GSE3837"

"chaperone ANP32A", not "chaperon ANP32A"

Dear editor, dear reviewers,

First, we want to thank everyone for the time invested and for carefully evaluating our article. We appreciate this effort and see that most suggestions will clearly improve our manuscript.

In the following, we will address the reviewer's comments point by point and comment on our respective changes:

Reviewer #1:

This study build on previous work from the same lab identifying CTBP2 function in transcriptional regulation of inflammation by glucocorticoids(Greulich et al, 2021). The response to glucocorticoids is complex. Many of the actions in macrophages are due to transcriptional induction of feedback regulators such as DUSP1. In macrophages GR binding sites are associated with binding sites for the macrophage-specific transcription factor, PU.1 and glucocorticoid action is associated with sustained chromatin decompaction (PMID: 29241532. PMID: 26663721). The current study is focused more on negative regulation of the response to lipolysaccharide (LPS). The authors should place this work more in the context of known feedback regulation of this response.

We thank the reviewer for this suggestion. We now added RT-qPCR data on *Dusp1* and *Gilz* expression in J774.1 wild type and CtBP KO cells in Figure 4 (Fig. 4E). Additionally the results are described in the text (lines 204 f.):

“Of note, the expression of anti-inflammatory mediators such as *Dusp1* and *Tsc22d3* is not lost in *Ctbp* dKO cells (Fig. S4E).”

Detailed analysis of the LPS response in human macrophages revealed a sequential cascade of transient induction and repression over a very extended time course (PMID: 28263993). It is likely that CTBP2 contributes to that feedback regulatory cascade.

We appreciate that the reviewer brought up this point. We added a time series comparing *I11a* and *Ccl22* expression following LPS challenge in J774.1 wild type and CtBP dKO cells to Figure S4 (Fig. S4D). This experiments shows that generally the amplitude of expression is affected by CtBP loss rather than the expression kinetics. This is discussed in the text (lines 201 ff.):

“Analyzing the expression kinetics of these genes in wild type and *Ctbp* dKO cells over an extended time course of 12 h showed that CtBP loss led to elevated inflammatory responses at every investigated time point, without altering expression kinetics (Fig S4D).”

The authors might also refer to the known roles of HDACs and the significant literature on pharmacological inhibitors in inflammation. (PMID: 21570914;PMID: 26900475). The putative targets of Ctbp2 repression overlap with the selective impact of HDAC inhibitors.

We appreciate this comment and incorporated a sentence about HDAC1 being involved in *IL12b* regulation in the text (lines 308 ff.).

“Additionally, the histone deacetylase 1 (HDAC1) has been reported to limit the expression of IL12B in HEK293T cells, potentially independent of the NuRD complex (Lu et al, 2005).”

Nevertheless, given that CTBP2 only interacted with HDAC1 and HDAC2 (NuRD associated HDACs), we prefer not to include data on gene expression following HDAC inhibition because we observed common HDAC inhibitors to lack specificity towards selected HDACs, especially among the class I HDAC inhibitors (see also: [https://www.cell.com/cell-reports/pdf/S2211-1247\(24\)00600-4.pdf](https://www.cell.com/cell-reports/pdf/S2211-1247(24)00600-4.pdf) for more details.)

Overall, the work is technically competent and convincing providing multiple lines of evidence to support the proposition that oligomeric CTBP2 represses a subset of LPS-inducible genes in macrophages.

It appears that results and discussion have been combined. I think that the story would be easier to follow if there were separate results and discussion sections and discussion was strictly removed from the results.

We thank the reviewer for this suggestion and agree that the article would be improved after separating the results from the discussion part. We now have two separate sections to first present our results and later discuss them in the context of literature.

One minor technical question. In the methods the bone marrow-derived macrophages are generated in a rather unconventional medium (to me at least) involving high concentrations of L929 conditioned medium and serum. The cells are then transferred to what is called macrophage serum free medium. This is not described in any detail and the product description on Thermo Fisher is rather nebulous. Following back through two citations, it appears that the medium contains 10ng/ml M-CSF as well as various additives. The authors should cite the original papers and clarify the composition.

We thank the reviewer for this comment. We reached out to Thermo Fisher to get more details about the medium. While we were ensured that the SFM does not contain M-CSF or related cytokines such as GM-CSF, the company keeps the detailed composition confidential.

In regards of the L929 supernatant for differentiation medium, we added a citation to our methods part that includes the used protocol (Barish et al. 2005 ;line 403). Generally, L929 supernatant has been characterized by mass spectrometry previously and its effects on BMDM differentiation has been compared to differentiation via recombinant M-CSF (PMID: 33853969).

Two references are incomplete and 2019 paper on BioRxiv (Li et al) is published PMID: 32174788

We thank the reviewer for carefully evaluating our references. We have changed the following references accordingly: Stankiewicz 2014, Li 2020, Cunningham 2019

Reviewer #2:

Strickland et al. provide a comprehensive study of the transcriptional corepressor role of the nuclear protein CTBP2 in macrophages. Strengths of the work include the use of multiple mechanistic levels of analysis (RNAseq, ChIPseq, ChIP-MS, nanoBRET) in cells depleted of CTBP2 or the related CTBP1 by RNAi or sgRNAs and reconstituted with wild-type or dimerisation/oligomerisation-deficient point mutants of CTBP2. All these data are finally integrated into a model showing that JUNB may act as a recruitment factor for CTBP2 to specific genomic loci in enhancers or promoters of prototypical pro-inflammatory LPS-induced genes such as Il1a or Ccl22, in order to repress gene expression in its multimerised form. In conclusion, this study provides compelling evidence for a bipartite recruitment mechanism of CTBP2, whereby TFs such as JUNB recruit the protein and its oligomerisation provides a platform for further recruitment of transcriptional repressor complexes. These findings represent interesting new information on negative gene regulatory mechanisms in the innate immune response. The manuscript is well written and the data are consistent with the author's conclusions.

Specific points:

Fig. 3D. Why was this experiment performed in murine fibroblasts? The author should add data from untreated cells to show whether the interaction of JunB with Ctbp2 is regulated by LPS. Immunoblots should be provided to show equal levels of fusion protein expression under all conditions.

Referee Cross-Comments:

I agree with most of the other reviewer's comments and suggest that it should be sufficient to address them by modifying the manuscript text and some of the data presentations accordingly.

I do not believe that these comments justify extensive additional experiments in this already large study.

We thank the reviewer for this comment. The reason for initially using NIH 3T3 cells, which are murine fibroblasts, is the high transfectability of these cells, which made establishing the assay easier. We have now replaced this piece of data by an assay performed in J774.1 cells, which are macrophage-like cells, to better support the storyline (Fig. 3D; lines 181 ff.). Anyways, we did not include a vehicle control because we already show the treatment dependency of the interactions in a naïve background using ChIP-MS, ChIP-seq and ChIP-qPCR. We have only performed this assay in order to demonstrate that the proteins JUN, JUNB and RELA have the potential to physically associate with CTBP2 in living cells.

Assessing equal protein expression in vehicle and LPS treated cells anyways was not feasible due to the lack of remaining antibodies. Moreover, given that transfection with FuGene and plasmid DNA can already pre-activate inflammatory gene expression in macrophages (PMID: 30084169), we would expect to have a weak vehicle control in the suggested experimental setting and feel like we have shown LPS dependency sufficiently using other assays.

Reviewer #3:

Strickland et al. study the roles of the the coregulators CTBP1/2 in mouse and human macrophage activation, find CTBP2 to be involved in inflammatory gene regulation but not CTBP1. CTBP2 binds to CREs along with NF- κ B and AP-1 and requires dimerization to limit transcription activation by TLR4 activation, likely via interactions with other repressor proteins. Finally, the authors show that a monomeric CTBP2 mutant continues to interact with JUNB and RNA Pol II and leads to further increased transcription activation by LPS.

Overall, the study provides a plethora of data to assess the contributions of CTBP1 and CTBP2 to gene activation by LPS in macrophages that will be interesting for the field. The manuscript would benefit from adjusting some of the language describing the findings and conclusions to match the data, additional ways of presenting the data, and expanding the discussion of prior data.

Major concerns:

1. Inaccurate nomenclature is confusing and sometimes misleading.

a) The authors show that in the absence of CTBP2, LPS-mediated induction of transcription of many genes is augmented. However, given that these genes are also highly transcriptionally activated in the presence of CTBP2 (10-1000-fold (e.g. PMID: 29403501)), it

would be more accurate to call CTBP2 a limiter of inflammatory gene activation throughout the manuscript, rather than "a repressor of inflammatory gene expression".

We thank the reviewer for this suggestion and we have changed the word "repress" by "control", "limit", "attenuate" or "mitigate" throughout the text. Nevertheless, we still term CTBP2 a co-repressor, because this term is used in related literature and describes the molecular function of a protein without direct DNA-binding activity involved in transcriptional repression.

b) The subtitle "CTBP2 binds JUNB and RELA motifs close to inflammatory genes in macrophages" inadvertently suggests that CTBP2 recognizes and directly binds DNA, which is not the case.

We agree with the reviewer and changed the subtitle in the text and in the respective figures to "CTBP2 occupies JUNB and RELA DNA motifs close to inflammatory genes in macrophages" (lines 122, 734, 751) to stress that this is not active DNA binding but rather a recruitment event involving a transcription factor. Moreover, we have avoided the term "binding site" for CTBP2 ChIP in this chapter and replaced it with "CTBP2 occupied site".

c) While KDM1A has been shown to demethylate some transcription factors (e.g. p53, HIF-1a), I am not aware that KDM1A was described to demethylate RELA in any of the papers cited by the authors: "KDM1A has been recognized as a repressor of inflammatory responses by demethylation of pro-inflammatory transcription factors such as RELA (Hanzu et al., 2013; Liu et al., 2024; Wang et al, 2018)"

We agree with the reviewer that this cannot be supported by the cited references. Concordantly, we have changed the sentence to "In particular, KDM1A has been recognized as a repressor of inflammatory responses by restricting DNA-binding of pro-inflammatory transcription factors such as RELA, which parallels our observation that JUNB DNA-binding is negatively influenced by CtBPs (Hanzu et al., 2013; Kim et al, 2018; Liu et al., 2024; Wang et al, 2018)". (lines 312 ff.)

2. Incorrect or overstated conclusions.

a) The authors' conclusion that JUNB recruits CTBP2 to repress or down-modulate transcription of LPS-stimulated genes (IFN1a, IL12b) in macrophages contradicts previous findings that JunB was required for high-level LPS-driven transcriptional activation of proinflammatory cytokines such as Il1b, Il12b and TNF (Fontana, JI 2015). The authors cite Fontana et al. for mentioning the role of JUNB in proliferation but do not discuss or reconcile these discrepancies, especially with regard to their interpretation that JUNB exerts "anti-

inflammatory actions". This disagreement with the literature would need to be addressed in the discussion.

We agree with the reviewer's suggestion and apologize for formulating this incorrectly. We wanted to state that, for specific genes, compared to JUN, JUNB is a rather weak activator of cytokine expression and is known to antagonize JUN activity in certain cases. Therefore, we have changed the formulation to:

"Whereas JUNB can compensate for JUN during mouse development, it has been reported that JUNB partially antagonizes transcriptional effects of JUN on cytokine production in fibroblasts, causing attenuated expression of specific inflammatory target genes (Passegué et al, 2002; Szabowski et al, 2000). This supports our observation that CTBP2 limits inflammatory gene expression of specific JUNB target genes suggesting that CTBP2 fine-tunes JUNB-dependent gene expression in macrophages" (lines 298 ff.)

Fontana et al have used macrophage specific deletion of JunB and shown that the lack of JunB reduces the activation of Il1b, Tnf and Il12b. However, in our study we focus on CTBP2 which is JunB recruited co-repressor that attenuates the activation of pro-inflammatory genes (among them Il1a, Il1b and Ccl22). We do not claim that JunB is not pro-inflammatory. In contrast, we observe that JunB binding to DNA showed a tendency to increase in CTBP dKO cells, which aligns with its pro-inflammatory function (Fig. 4E). We also show that JunB interacts with CTBP2 (mono- and oligomers) downstream (Fig. 3A/D, Fig. 5B/E) and that only the oligomeric form of CTBP2 (which recruits co-repressors to the JUNB binding site) is able to suppress the expression of Il1a which is over-activated otherwise due to an unopposed pro-inflammatory function of JUNB (possibly via co-activation). Those results clearly position CTBP2 downstream of JUNB-mediated pro-inflammatory gene regulation.

To stress this even more, we frequently termed JUNB a "pro-inflammatory transcription factor" throughout the text (e.g. lines 305, 313, 316) and state that inflammatory gene expression is JUNB dependent (line 299)

b) Text incongruent with data: Fig. 1E shows CCL22 not significantly increased with CTBP2 KD, and for sure not "hyper-activated", and none of the changes in IL12B mRNA with CTBP2 KD in Fig. S1E were statistically significant ($p = 0.1$). However, the manuscript states: "Indeed, knockdown of CTBP2 but not CTBP1 led to hyper-activation of CCL22 and IL12B following LPS treatment (Fig. 1E, Fig. S1E)." - this conclusion is not adequate and should be corrected to reflect the presented data.

We agree with the reviewer and have changed this statement accordingly: "Indeed, knockdown of CTBP2 but not CTBP1 led to a strong trend of elevated CCL22 and IL12B expression following LPS treatment (Fig. 1E, Fig. S1F). However, regulation of IL1A was not

dependent on CTBP2 alone, potentially indicating redundant roles of CTBP1 and CTBP2 in human monocyte-derived macrophages” (lines 118 ff.)

c) *While the authors show that JUNB and CTBP2 interact, they do not show that e.g. CTBP2 fails to be recruited in the absence of JUNB, e.g. following JUNB KO. In the absence of this data, conclusions along the lines that "CTBP2 is recruited ... by JUNB" should probably be weakened to "Our data strongly indicates that CTBP2 is recruited ... by JUNB".*

We sincerely thank the reviewer for bringing up this concern. While we have strong indication that JUNB recruits CTBP2 to its target genes, indeed we are lacking experimental proof for this statement. Therefore, we have changed it to:

“Motif enrichment analysis at differentially occupied loci demonstrated that CTBP2 occupancy was gained at RELA and JUNB DNA motifs in response to LPS, suggesting that these transcription factors may recruit CTBP2 to the respective sites.” (lines 142 ff.)

“We suggest that CTBP2 acts as a scaffold protein that brings co-repressors to specific genomic loci via its interaction with pro-inflammatory transcription factors such as NF- κ B and JUNB” (lines 304 ff.)

d) *The manuscript tries to arrive at significant conclusions based on often very small quantitative differences of uncertain significance. For example, the conclusion that CTBP2 knockout increases JUNB binding is based on very small differences in enrichment measured by ChIP-qPCR (Fig. 4E) that are interpreted to reflect significant changes in JUNB binding to different genomic locations. However, the description of the experimental methodology and the significance calculation need to be improved. Specifically, the %input measurement used for t-test is a derived measurement that incorporates both a IP signal and an input signal. These independent variables are encumbered by individual measurement errors that contribute to the overall error in the derived %input measurement, which therefore is larger than the individual measurement errors of each value. However, whether and how the errors were propagated is not described. Further, it is not clear whether the values shown are from independent experiments with different KO cell populations (i.e. biological replicates) or technical ChIP replicates with the same cell population, or technical PCR replicates from the same ChIP. Together with the minute reported differences in ChIP-qPCR signal between different cell clones/populations (equivalent to less than 0.5 cycles in qPCR), this puts into question the significance of the small ChIP-qPCR differences and weakens the conclusion.*

We thank the reviewer for carefully assessing our experimental description. The experiment was performed with three independent cell populations from the same knockout or wild type

clone and two technical IP replicates per population. We have tried to clarify our setup and data analysis in more detail:

“Cq-values for ChIP samples and respective inputs were assessed via qPCR in technical triplicates. The mean of technical triplicates was normalized to the respective input in percent. The mean percent input of three independent experiments (with two technical replicates each) was plotted as bar with the standard deviation as error bar and single dots as individual replicates“ (lines 532 ff.)

e) The conclusion: "In contrast to the CTBP2 cistrome in murine livers (Sekiya et al., 2021), we find that in macrophages CTBP2's cistrome is enriched for motifs of inflammatory transcription factors indicating that CTBP2 binding is highly tissue-specific.", is based on misinterpretation of the data and should be corrected. For example, the fact that the subset of LPS-inducible CTBP2 binding locations are enriched for motifs of LPS-activated TFs NF- κ B and AP-1, while CTBP2 binding locations in unstimulated liver cells are not, does not mean that CTBP2 binding is tissue-specific. To make that statement, the authors would need to compare motif enrichment in either LPS-inducible macrophage-specific CTBP2 peaks to the inducible peaks in liver cells similarly stimulated with LPS, or compare the motifs associated with the basal CTBP2 binding patterns in macrophages (~20k sites) and liver cells.

We thank the reviewer for noting this and agree that the conclusion is not valid based on our data presentation. We therefore have removed this statement from the manuscript.

Minor concerns and suggestions:

The ChIP-MS protocol is not provided, and the corresponding citation in the Methods section is missing ("Samples were then shipped to the proteomics research infrastructure Copenhagen and subjected to their pipeline to remove remaining nucleic acids and generate peptides for mass spectrometry as described previously (REF)."). Due to the missing protocol, it is not clear how/whether the samples were DNase-treated to only measure direct protein-protein interactions.

We thank the reviewer for carefully assessing our methods part. We have added the experimental description and the PRIDE accession numbers to the manuscript. (lines 387 ff., 569 ff.)

The DNA binding motifs for c-Jun and JunB are the same, therefore the statement that motif differences correlate with differences in binding is not valid. ("Contrarily, we did not observe

binding of NFKB1 or JUN at any investigated site, which is in line with the motif enrichment results from our ChIP-seq experiments (Fig. 2D).")

We agree with the reviewer and have changed the statement from the manuscript to "We did not observe binding of NFKB1 nor JUN at any investigated site (Fig. 3E)" (lines 193 f.)

From the way the RNA-seq data in Fig. 1/S1 is presented, as a z-scored heat map of relative expression values of the most significantly changing genes with and without LPS treatment, or a volcano plot of differences in LPS-induced relative expression values, it is difficult to verify the conclusion that CTBPs repress transcription, as it is impossible to gauge the absolute gene expression differences induced by LPS in the presence or absence of CTBP1/2. A supplemental figure showing e.g. a MA-style plots of averaged vehicle vs. LPS treatment RNA-seq data for wt, CTBP1 and CTBP2 KD would enable the reader to get a better picture of the magnitude differences in absolute transcript levels caused by disrupting CTBP1/2.

We thank the reviewer for pointing out this concern regarding the impact of CtBP knockdown on the total LPS response. We understand that this request is supposed to clarify whether the massive transcriptional induction of LPS target genes is affected by our genetic disturbances. We now provide the desired MA-plots for LPS over vehicle gene expression in each genotype in Fig S1D and added a sentence to the text:

"Of note, the majority of LPS-responsive genes were not affected by knockdown of Ctbp1 or Ctbp2, indicating a specific role of Ctbp2 in fine-tuning a subset of inflammatory genes upon LPS challenge (Fig. S1D)." (lines 108 ff.)

The authors could use NanoBRET to show that CTBP2 does not interact with Jun.

We thank the reviewer for this suggestion. Our ChIP-MS data indicates that JUN and CTBP2 can interact, however this interaction seemed to be specific to vehicle conditions in primary macrophages. In light of the fact that Nano-BRET uses overexpressed proteins to proof the interaction-potential of two proteins e.g. JUN and CTBP2, we think that the suggested experiment might not be able to disentangle this signal-specific interaction. Anyways we now provide data that JUN and CTBP2 can weakly interact in LPS conditions, however to a much lower extent than JUNB (Fig. 3D). Moreover we added "Bioluminescence resonance energy transfer from Nano-luciferase-labeled RELA, Nano-luciferase-labeled JUNB or Nano-luciferase-labeled JUN to fluorescently labeled CTBP2-Halo-tag was measured (Fig. 3D). While this demonstrates the ability of CTBP2 to physically interact with these transcription factors, JUNB outperformed RELA and JUN, indicating that JUNB might have a higher affinity for CTBP2 compared to RELA and JUN" to the text (lines 183 ff.)

Jun and NFKB1 ChIP-qPCRs should include positive controls.

We agree with the reviewer that a positive control would highlight the validity of this assay, however in the absence of a known positive locus we have not included this in our experiment. If the reviewer insists, we can offer to remove the ChIP-qPCR data for JUN and NFKB1 from the figure.

Typos:

"RAW264.7", not "Raw246.7"

"GSE38379", not "GSE3837"

"chaperone ANP32A", not "chaperon ANP32A"

We are grateful to the reviewer for pointing out these typos. We have changed them accordingly in the text or removed the sentence containing it (RAW264.7). (lines 146, 254).

Again, we want to sincerely thank all reviewers for carefully assessing our manuscript and giving suggestions for improvements.

May 20, 2025

RE: Life Science Alliance Manuscript #LSA-2025-03263-TR

Prof. Nina Henriette Uhlenhaut
TUM School of Life Sciences Weihenstephan
Metabolic Programming
Gregor-Mendel-Str. 2
Freising 85354
Germany

Dear Dr. Uhlenhaut,

Thank you for submitting your revised manuscript entitled "C-terminal binding protein 2 interacts with JUNB to control macrophage inflammation". Upon checking the revised manuscript and rebuttal letter, and in view of the strong support from reviewers on the initial submission, we determined that further reviewer input was not necessary. We would be happy to publish your paper in Life Science Alliance pending final revisions necessary to meet our formatting guidelines.

- Please add ORCID ID for corresponding author--you should have received instructions on how to do so.
- Please add the X and Bluesky handles of your host institute/organization as well as your own or/and one of the authors in our system.
- Please remove Orcid IDs from the names of the authors on the title page.
- On the title page of the manuscript, please provide the full name of each author, including middle names as initials, formatted as follows: First name, middle initial, Last name.
- Please remove the word count provided after the Summary Blurb, Abstract, etc.
- Please consult our manuscript preparation guidelines <https://www.life-science-alliance.org/manuscript-prep> and make sure your manuscript sections are in the correct order
- The contributions selected for Nina Henriette Uhlenhaut do not qualify them for authorship. Please either update the contributions in our system and the Author Contributions section of the manuscript, or let us know if the author needs to be removed (and added eventually to the acknowledgment section).
- Please add your main, supplementary figure, and table legends to the main manuscript text after the references section.
- Please separate the Figure legends and Supplemental Figure legends into separate sections.
- We encourage you to revise the figure legends for Figure S1 such that the figure panels are introduced in alphabetical order.
- Please add a callout for Table S4 to your main manuscript text.
- Please add molecular weights to the blots in Figures 1A, S4B, and S5B.

A. FINAL FILES:

B. MANUSCRIPT ORGANIZATION AND FORMATTING:

Sincerely,

May 26, 2025

RE: Life Science Alliance Manuscript #LSA-2025-03263-TRR

Prof. Nina Henriette Uhlenhaut
Technical University of Munich
Gregor-Mendel-Str 2
Freising 85354
Germany

Dear Dr. Uhlenhaut,

Thank you for submitting your Research Article entitled "C-terminal binding protein 2 interacts with JUNB to control macrophage inflammation". It is a pleasure to let you know that your manuscript is now accepted for publication in Life Science Alliance. Congratulations on this interesting work.

DISTRIBUTION OF MATERIALS:

Again, congratulations on a very nice paper. I hope you found the review process to be constructive and are pleased with how the manuscript was handled editorially. We look forward to future exciting submissions from your lab.

Sincerely,
